# Vertical depletion of ophiolitic mantle reflects melt focusing and interaction in sub-spreading-center asthenosphere

Qing Xiong [1,2] ✉, Hong-Kun Dai [1,2], Jian-Ping Zheng [1] ✉, William L. Griffin [2], Hong-Da Zheng [1], Li Wang[1] & Suzanne Y. O'Reilly [2]

Decompressional melting of asthenosphere under spreading centers has been accepted to produce oceanic lithospheric mantle with vertical compositional variations, but these gradients are much smaller than those observed from ophiolites, which clearly require additional causes. Here we conduct high-density sampling and whole-rock and mineral analyses of peridotites across a Tibetan ophiolitic mantle section (~2 km thick), which shows a primary upward depletion (~12% difference) and local more-depleted anomalies. Thermodynamic modeling demonstrates that these features cannot be produced by decompressional melting or proportional compression of residual mantle, but can be explained by melt-peridotite reaction with lateral melt/rock ratio variations in an upwelling asthenospheric column, producing stronger depletion in the melt-focusing center and local zones. This column splits symmetrically and flows to become the horizontal uppermost lithospheric mantle, characterized by upward depletion and local anomalies. This model provides insights into melt extraction and uppermost-mantle origin beneath spreading centers with high melt fluxes.

Asthenosphere upwelling, lithosphere generation, and plate divergence in oceanic spreading centers (such as mid-ocean ridges and forearc/backarc centers) drive the dynamics of plate tectonics and regulate the cycling of mass and heat between Earth's interior and at least two-thirds of Earth's surface[1,2]. The physicochemical processes (e.g., asthenospheric flow, partial melting, melt migration, and melt-rock interaction[3–7]) and parameters (e.g., mantle potential temperature, spreading and upwelling rates, mantle source composition, plate-lid thickness[8–11]) under spreading centers have produced oceanic lithosphere of great complexity in structure and composition[12,13].

Oceanic lithospheric mantle with complex characteristics and origins has been sampled from the present-day ocean floor and fossil ophiolites[1]. The abyssal peridotites (commonly serpentinized) have been dredged and drilled from the tectonically exposed shallowest mantle in mid-ocean ridges or along transform faults[12–15]; the deepest

reported drill hole into the mantle is only ~200 m at the Mid-Atlantic Ridge[16]. The lack of deeper sampling of a complete vertical section of oceanic lithospheric mantle limits our understanding of the formation and evolution of oceanic lithospheric mantle and the mantle dynamic processes beneath spreading centers.

Fortunately, appropriate ophiolites with excellent mantle-rock exposure and limited modification in orogenic belts can provide direct and complete snapshots of oceanic lithospheric mantle with clear spatial context[1]. They are an indispensable source of information on the origins of oceanic-mantle heterogeneity and the petrochemical and dynamic processes under oceanic spreading centers[17–24]. For example, the fractal dunite melt-channel system in the upwelling residual mantle has been proposed by studies of ophiolites (mainly the Oman example) to illustrate melt-extraction processes and mantle dynamics under oceanic spreading centers[6,25].

[1]State Key Laboratory of Geological Processes and Mineral Resources, School of Earth Sciences, China University of Geosciences, Wuhan 430074, China. [2]Australian Research Council Centre of Excellence for Core to Crust Fluid Systems (CCFS) and GEMOC, School of Natural Sciences, Macquarie University, Sydney, NSW 2109, Australia. ✉e-mail: xiongqing@cug.edu.cn; jpzheng@cug.edu.cn

In this study, we have selected the well-exposed Kangjinla ultramafic massif (eastern part of the Luobusa ophiolite) in the Yarlung Zangbo suture zone (South Tibet; Fig. 1), which displays the vertical architecture of a ~2-km-deep uppermost lithospheric mantle section in the Neo-Tethyan Ocean[20,26,27]. We have carried out systematic high-density sampling, detailed petrographic investigations, and geochemical analyses of whole-rock and mineral compositions, as well as thermodynamic modeling of relevant decompressional melting and melt-peridotite interaction processes. Our aims are to provide a high-resolution view of the lithological and compositional variations in a section of oceanic lithospheric mantle, and to reveal the dynamic processes responsible for the compositional features of uppermost lithospheric mantle under oceanic spreading centers.

## Results and discussion
### The Kangjinla ophiolitic mantle

The ophiolites in the ~2000-km-long Yarlung Zangbo (YZ) suture (South Tibet; Fig. 1a) represent relics of oceanic lithosphere formed at various spreading centers in the Neo-Tethyan Ocean[23,24,26–30], which separated Greater India in the south from the Lhasa block in the north during the Mesozoic[31]. The YZ ophiolites expose tens to hundreds of km² outcrops of individual bodies; from east to west the main bodies are the Luobusa, Zedang, Xigaze, Saga, Dangqiong, Xiugugabu, Purang, and Dongbo ophiolites (Fig. 1b). The Luobusa ophiolite is the most

well-known because it contains the largest chromitite ore deposits in China[32] as well as peculiar ultrahigh-pressure and super-reduced minerals identified from the mantle rocks[33]. Contrasting tectonic origins proposed for the Luobusa ophiolite would have it produced in (i) a mid-ocean ridge[26,35], (ii) primarily a mid-ocean ridge overprinted by subduction-zone modification[20,32], (iii) a single-stage nascent forearc[27], or (iv) multiple episodes of subduction-zone cycling[36]. However, there is a consensus that the major architecture of the Luobusa ophiolite was generated in an oceanic spreading center controlled by plate divergence and asthenospheric upwelling.

The Luobusa ophiolite is a south-dipping tectonic slice (Fig. 1c) with an exposed length of ~42 km and width of ~1–3 km as well as a geophysically constrained thickness of ~2–3 km[37]. It is sandwiched between northern Eocene molasse (Luobusa Formation) at the base and the southern Triassic flysch (Langjiexue Group) on top[20]. On the outcrop, from north to south, the ophiolitic sequence includes a serpentinite mélange zone enclosing mafic-ultramafic cumulate lenses, a paleo-Moho transition zone of dunite enclosing harzburgite relics, and a gradation from clinopyroxene-poor harzburgite, through clinopyroxene-rich harzburgite to lherzolite. This zonation reveals that the mantle section was overturned during its emplacement[20].

From west to east, three segments (Luobusa, Xiangkashan, and Kangjinla) can be further subdivided, and they are continuous in the E-W direction. Previous investigations have shown that the three

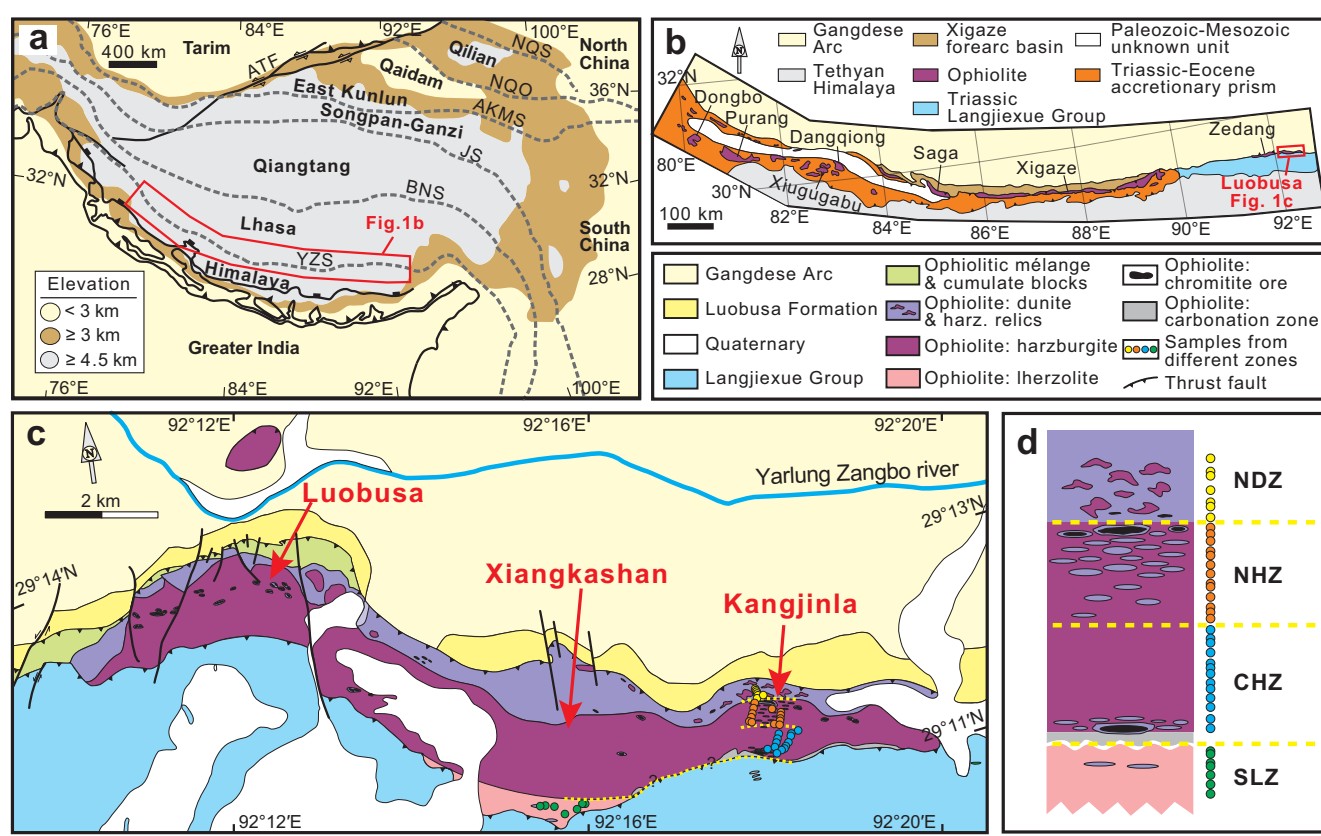

**Fig. 1 | Simplified tectonic and geological maps showing the Kangjinla ophiolitic mantle in the Yarlung Zangbo Suture (South Tibet).** Major tectonic units of the Himalayan-Tibetan orogenic system (**a**; modified from DeCelles et al.[67]), geological sketch map of South Tibet showing the Yarlung Zangbo Suture and major ophiolites (**b**; modified from Dai et al.[68]), and the simplified geological map illustrating the Luobusa ophiolite and adjacent tectonic units (**c**; modified from Liang et al.[69]). This study focused on the Kangjinla segment and the southernmost portion of the Xiangkashan segment, both of which display intact occurrence, direct exposure, and minimum alteration. We collected representative peridotites by hammers on the well-exposed outcrops with sampling spacing of several to tens of meters. The studied segments were then reconstructed as four zones (**d**), from north to south including the northern dunite zone (NDZ), northern harzburgite zone (NHZ), central harzburgite zone (CHZ) and southern lherzolite zone (SLZ). The four zones can represent the major architecture of the ophiolitic mantle, and construct a mantle profile defined as "the Kangjinla ophiolitic mantle" in this study (**d**). Yellow dashed curves in **c** and **d** mark the suggested boundaries between the four zones, and circles with different colors show the sampling positions. YZS Yarlung Zangbo Suture, BNS Bangong-Nujiang Suture, JS Jinsha Suture, AKMS Anyimaqen-Kunlun-Muztagh Suture, NQO North Qaidam Orogen, NQS North Qilian Suture, ATF Altyn Tagh Fault.

segments have similar internal architecture[20,27,38–40], and the primary lithospheric mantle stratigraphy is completely exposed in the Kangjinla segment and the southern part of the Xiangkashan segment (Fig. 1c). We therefore chose these two segments for high-density sampling, aiming to cover the whole mantle section of the Luobusa ophiolite.

The studied segments were then reconstructed as a complete mantle profile defined as "the Kangjinla ophiolitic mantle", which can be subdivided into four zones based on lithological associations and petrographic features (Fig. 1d; Supplementary Dataset 1). From north to south (top to bottom of the reconstructed stratigraphy), they are the northern dunite zone (NDZ), northern harzburgite zone (NHZ), central harzburgite zone (CHZ) and southern lherzolite zone (SLZ). They form a continuous mantle section ~2 km wide in the N-S direction on the outcrop, reflecting an estimated maximum thickness for the Kangjinla ophiolitic mantle of ~2 km according to field and geophysical observations[20,37]. At the contact between the CHZ and the overlying Triassic black slates, the peridotites are strongly carbonated and were not sampled in this study (Fig. 1c, d). This mantle section thus represents a snapshot of the uppermost lithospheric mantle in the Neo-Tethyan Ocean[20,27,38–40].

## Sample descriptions

In the reconstructed Kangjinla profile, rare thin chromitite veins grew in the SLZ, small chromitite pods are found in the basal CHZ, and large chromitite ore bodies occur in the upper NHZ and the NDZ (Fig. 1d). Dunites are closely associated with and enclose the chromitites, and increase in size from the SLZ to the NDZ (Fig. 1c, d). The NDZ is made up not only of dunites but also abundant harzburgite relics. We collected forty representative peridotites (mainly harzburgites, some lherzolites and rare pyroxene-bearing dunites) from the four zones, to fully cover the Kangjinla lithospheric mantle profile (Supplementary Fig. 1a). The sampled peridotites have primary mineral assemblages of olivine (Ol) + orthopyroxene (Opx) + spinel (Sp) ± clinopyroxene (Cpx) ± sulfide (Sulf), and show porphyroblastic textures and plastic deformation (Supplementary Fig. 1b–m). Petrographic variations of the peridotites from the SLZ to the NDZ are systematic (Supplementary Dataset 1), including gradual upward disappearance of Cpx and Sulf, decrease in modal Opx and increase in modal Ol (Supplementary Figs. 1, 2).

The SLZ peridotites are mainly lherzolites (with ~5.7–10.2% Cpx as porphyroblasts), contain Fe-Ni sulfides (mainly pentlandite, ~0.05–0.1%), and show higher abundance of pyroxene (reaching maximum of ~40%) than those of other zones, except for a few Cpx-free harzburgites close to dunite lenses (e.g., sample KJL14-05C). Lherzolite KJL14-05A from the SLZ contains the most abundant Opx and Cpx (mainly as porphyroblasts) and the least olivine in the forty samples (Supplementary Fig. 2). The Cpx porphyroblasts are largest in size (reaching ~2–3 mm in diameter) within the studied samples, and show irregular shapes, undulatory extinction, and exsolved Opx laths (Supplementary Fig. 1k–m). These petrographic features resemble those of residual lherzolites derived mainly from partial melting of asthenosphere[23,24]. In the CHZ and NHZ, the peridotites generally become more pyroxene-poor and Ol-rich from south to north, with gradual reduction in the grain sizes of both Opx and Cpx and enlargement of Ol grains (Supplementary Figs. 1e–j, 2). Some pyroxene-bearing dunites from the basal CHZ (e.g., samples 18KJL09-01 and 16KJL20-01) are direct wall-rocks of typical dunite lenses, and contain resorbed Opx and rare or absent Cpx, similar to the Cpx-free harzburgite KJL14-05C in the SLZ (Supplementary Dataset 1). In the NDZ, the harzburgites occur as relict enclaves with diffuse boundaries within the dunites, and show the most abundant Ol, the least modal Opx and rare Cpx. The spinels form trails and become rounded or euhedral, similar to those in the dunites (Supplementary Fig. 1b–d). Sample KJL1522-04 shows higher modal Opx than other NDZ harzburgites.

## Elemental compositions of whole rocks and minerals

Whole-rock major- and trace-element compositions of the Kangjinla peridotites are highly variable (Supplementary Datasets 2, 3), covering two-thirds of the global range of abyssal peridotites (Figs. 2, 3). Large compositional ranges are also shown in Sp, Cpx, Opx, and Ol (Supplementary Datasets 4–9). The variations between LOI and whole-rock/mineral elemental concentrations suggest that secondary fluid metasomatism (including serpentinization) has only enriched fluid-mobile elements but did not affect the compositions of major oxides (e.g., $MgO$, $FeO_T$, $SiO_2$, $Al_2O_3$, and $CaO$) and many incompatible elements (e.g., Ti, Yb; Supplementary Figs. 3, 4). To reveal the high-temperature mantle processes under oceanic spreading centers, we therefore only consider fluid-unaffected major elements and intermediately-slightly incompatible trace elements in whole rocks and minerals.

Samples KJL14-05C, 16KJL20-01, and 18KJL09-01 display local abnormal depletion (Figs. 2, 3), and sample KJL1522-04 shows lower $MgO$ and $FeO_T$ and higher $SiO_2$ which reflect metasomatic addition of Opx. All the other peridotites show gradual variations from the SLZ to the NDZ for major oxides and intermediately-slightly incompatible trace elements in both whole rocks and minerals (Figs. 2a–l, 3a–d; Supplementary Figs. 4–7). For example, from the bottom to the top of the Kangjinla mantle section, whole-rock $MgO$ (Fig. 2a), Cr# ($Cr^{3+}/(Cr^{3+}+Al^{3+})$) for Sp (Fig. 2c) and Opx (Fig. 2e), and Ol Mg# ($Mg^{2+}/(Mg^{2+}+Fe^{2+})$; Fig. 2f) gradually increase, while whole-rock $Al_2O_3$ (Fig. 2b) and Cpx $Al_2O_3$ (Fig. 2d), as well as Ti and Yb in both whole rocks and pyroxenes (Fig. 2g–l) decrease upwards.

Among the four zones, whole-rock $MgO$ contents show strong negative correlations with $SiO_2$ (Fig. 3a), $CaO$ (Fig. 3c), and $Al_2O_3$ (not shown), and positive co-variations with $FeO_T$ (Fig. 3b). Whole-rock Ti and Yb show a clear positive correlation (Fig. 3d). These linear variations are also seen in mineral Mg#, Cr# and $Al_2O_3$ contents from the SLZ to the NDZ (Supplementary Fig. 5). In addition, all whole rocks, Cpx and Opx exhibit consistent left-leaning, depleted REE patterns (Supplementary Fig. 6), as well as "U-shaped" multi-element patterns with strong enrichments in fluid-mobile elements (e.g., Cs, Rb, Ba, Pb, Li; Supplementary Fig. 7). The fluid-mobile elements and some light REEs (e.g., La) share similar enrichment extents across the four zones, and show positive correlations with LOI contents (Supplementary Figs. 3, 6, 7), implying overprinting by fluid metasomatism during later subduction-zone or orogenic modifications after the formation of the Kangjinla mantle section in a spreading center[20,27,32].

## Compositional variations of oceanic uppermost mantle represented by a Tibetan ophiolite

Ophiolites have been widely interpreted as relics of juvenile oceanic lithosphere produced in spreading centers (mid-ocean ridges and forearc/backarc centers)[1,19]. For example, the Oman ophiolite formed in a fast-spreading center within a time span of ~1 Ma, and was obducted soon after leaving the spreading-center regime[41]. This means that the main architecture of ophiolites can largely record the birth and infancy of oceanic lithosphere, but cannot document the later thickening and accretion after the lithosphere moving away from the spreading center, as proposed by the age-related half-space cooling model or plate model[9,10,42,43]. Therefore, the detailed observations of the Kangjinla ophiolite from this study can provide a direct close-up view of juvenile oceanic lithospheric mantle (particularly of its ~2-km-deep uppermost portion), and can reflect the thermal and dynamic processes from asthenosphere to lithosphere beneath oceanic spreading centers.

The high-density sampling and systematic investigations of the reconstructed Kangjinla ophiolitic profile show a first-order lithological zoning in the lherzolitic-harzburgitic mantle framework, with generally increasing olivine but decreasing pyroxenes from bottom to top (Fig. 1d; Supplementary Figs. 1, 2; Supplementary Dataset 1). The petrographic features are consistent with the gradual variations in

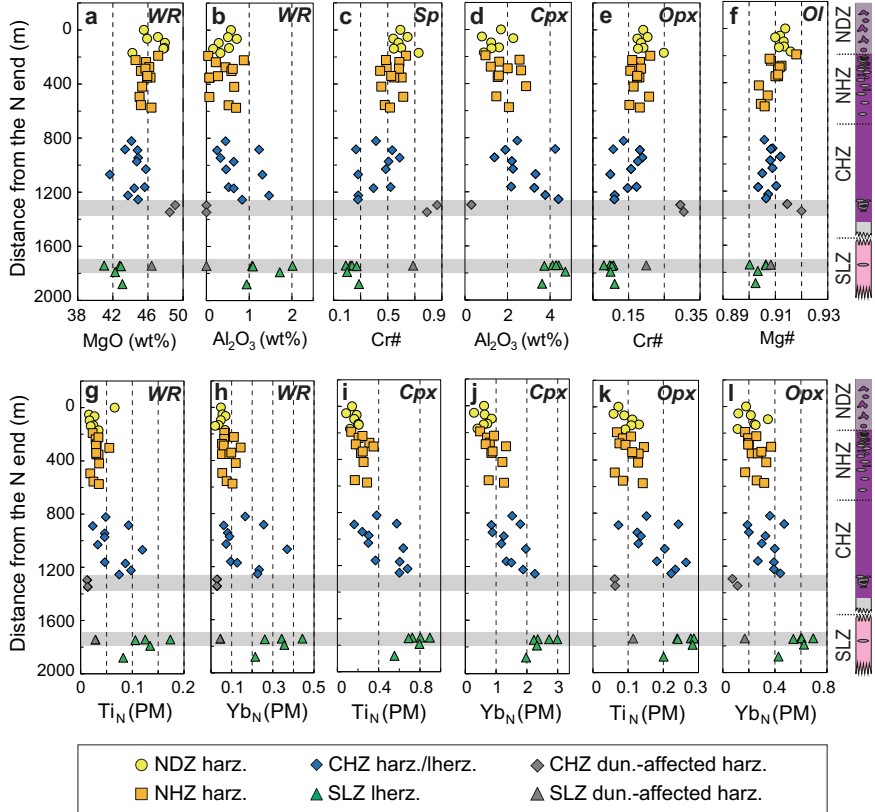

**Fig. 2 | Compositional variations of the Kangjinla ophiolitic mantle section (South Tibet).** North-south variations of major- (**a**–**f**) and trace-element (**g**–**l**) compositions for whole rocks and minerals of peridotites from the four zones of the Kangjinla ophiolitic mantle. The distance of each sample was calculated by transformation of each longitude relative to the north end as 0 m. Petrological columns were shown at the right sides to illustrate the N-S lithological variations and zonation of the Kangjinla ophiolitic mantle, with the same legends as those in Fig. 1. **a** Whole-rock anhydrous MgO (wt%), **b** whole-rock anhydrous $Al_2O_3$ (wt%),

**c** spinel Cr# ($Cr^{3+}/(Cr^{3+}+Al^{3+})$), **d** clinopyroxene $Al_2O_3$ (wt%), **e** orthopyroxene Cr#, **f** olivine Mg# ($Mg^{2+}/(Mg^{2+}+Fe^{2+})$), **g** whole-rock $Ti_N$ (PM, normalized to primitive mantle[70]), **h** whole-rock $Yb_N$ (PM), **i** clinopyroxene $Ti_N$ (PM), **j** clinopyroxene $Yb_N$ (PM), **k** orthopyroxene $Ti_N$ (PM), and **l** orthopyroxene $Yb_N$ (PM). Two gray bands mark the local abnormally-depleted zones affected by melt-peridotite interaction during dunitization. WR, whole rock; Sp, spinel; Ol, olivine; Opx, orthopyroxene; Cpx, clinopyroxene; harz., harzburgite; lherz., lherzolite; dun., dunitization.

whole-rock and mineral compositions (Figs. 2–4), suggesting a primary upward removal of chemical components incompatible with the mantle-melting residues. In the Kangjinla lower zones (SLZ and CHZ), some Cpx-free harzburgites or pyroxene-bearing dunites close to dunite lenses exhibit consumption of pyroxenes, addition of olivine and much stronger depletion than that of the wall-rock peridotites (Fig. 2; Supplementary Fig. 2). This local compositional depletion has been interpreted as the results of dunitization of lherzolites/harzburgites by interaction with silica-undersaturated silicate melts during melt migration via high-porosity channels[6,20].

Comparable compositional features of oceanic lithospheric mantle have also been observed in other ophiolites, such as Oman[18,21], Troodos[17], and Bay of Islands[44], but all with much lower sampling densities (hundreds to thousands of meters). In summary, gradual vertical depletion of ophiolitic mantle and local depletion anomalies are the prevailing and first-order phenomena in oceanic uppermost lithospheric mantle, if the sampling density of the ophiolites is high enough or the sampling depth of oceanic drill holes is deep enough to reveal the mantle compositional trends.

**Decompressional melting cannot produce the vertical depletion of oceanic uppermost mantle**
Under an oceanic spreading center, adiabatic upwelling and advection of asthenospheric mantle into the space produced by divergent spreading of the overlying plates shape the dynamic flow field of solid mantle underneath spreading centers[3,4,7,11,45]. The concomitant decompressional melting and melt extraction result in the focusing of

melts towards the axis of the spreading center and an interplay between melt and the upwelling mantle[6,25,46–48]. These processes, combined with variations in mantle potential temperature, spreading and upwelling rates, mantle source composition, asthenospheric flow patterns, plate-lid thickness, and melting mechanism have been proposed to produce the great compositional complexity in the mantle and crust of oceanic lithosphere[3–5,7–11]. The lithospheric mantle variations are best reflected by the first-order upward compositional depletion of residual mantle columns, which have been interpreted as the product of decompressional melting and lateral transposition as the compositionally stratified lithospheric mantle[4].

In this study, in order to reproduce the compositional variations of residual mantle columns under oceanic spreading centers, we modeled the isentropic decompressional fractional melting of a depleted-MORB-mantle (DMM) source[49] with assumed potential temperatures (Tp) of 1300 °C, 1350 °C, 1400 °C, and 1450 °C[50], using the pMELTS version[51] of the alphaMELTS 1.9 software package[52] (details see "Methods", Supplementary Datasets 10, 11). Because the melting of volatile- and/or pyroxenite-rich sources will start at deeper levels and plays an important role in the low-melting-degree situations and melt compositions[53,54], we only consider the melting of an anhydrous peridotitic source, which produces the major architecture of oceanic lithospheric mantle[11]. Moreover, ancient ultra-depleted components have been proposed to exist in the asthenosphere and oceanic lithospheric mantle[22,55,56], but their size, distribution, and relations to high solidus temperatures are still poorly known, leading to uncertainties about their distribution in the residual mantle columns and their

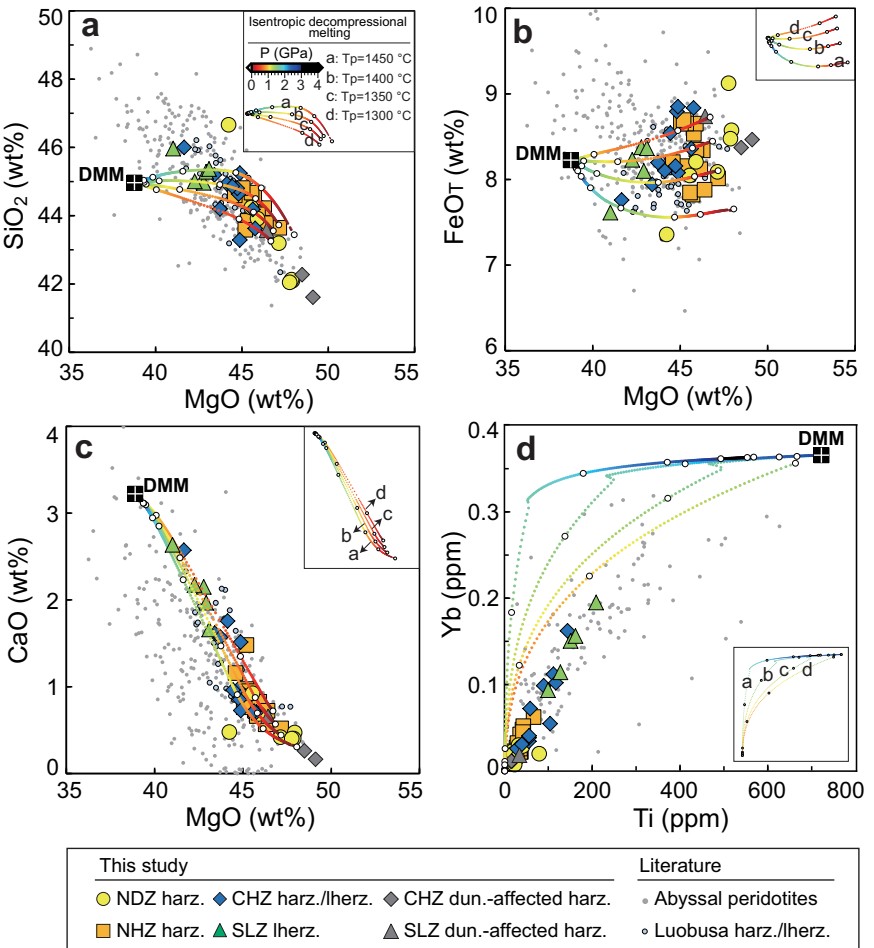

**Fig. 3 | Whole-rock compositional variations of the Kangjinla ophiolitic peridotites and decompressional melting modeling.** Variations of whole-rock MgO (wt%) versus SiO₂ (wt%, **a**), FeO$_T$ (wt%, **b**), and CaO (wt%, **c**), as well as whole-rock Ti (ppm) versus Yb (ppm, **d**) for the Kangjinla ophiolitic peridotites. Isentropic decompressional fractional melting trends (color-coded) for residual peridotites from a depleted-MORB-mantle (DMM) source[49] were modeled using the pMELTS version[51] of alphaMELTS 1.9 program[52], with mantle potential temperatures of 1300 °C, 1350 °C, 1400 °C, and 1450 °C. The color-coded pressure-decreasing gradient is of 0.1 kbar, and the white circles mark the pressure steps of 0, 0.5, 1.0, 1.5, 2.0, 2.5, 3.0, 3.5, and 4.0 GPa. The detailed melting conditions were listed in Supplementary Dataset 10. Small gray circles represent global abyssal peridotites without those veined by gabbro, pyroxenite, and dunite[13–15]; light-blue circles represent the previously reported harzburgites and lherzolites with unclear spatial contexts from the Luobusa ophiolite[20, 32, 35, 38].

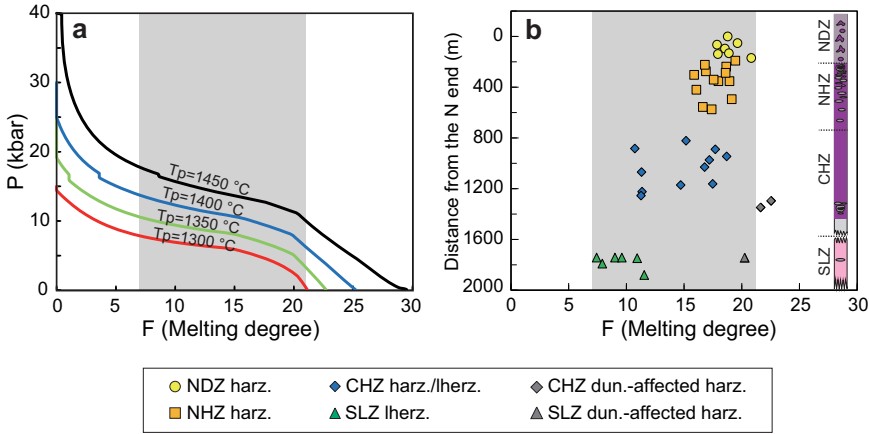

**Fig. 4 | Mantle thickness comparison between results from decompressional melting modeling and observations from the Kangjinla ophiolitic section.** Variations of partial melting degrees (F) versus pressure (P, kbar) for the isentropic decompressional melting residues (**a**), and F versus distance from the north end (m) for the Kangjinla peridotites (**b**). F in **a** was acquired by thermodynamic modeling as shown in Fig. 3, while F in **b** was calculated using spinel Cr# and the updated equation (F = 9*ln(Cr#)+23)[13]. Petrological column shown in **b** is the same as those shown in Fig. 1d. Gray zones in **a** and **b** mark the mantle "depletion" extents of the Kangjinla peridotites, and indicate the required pressure differences produced by the isentropic decompressional fractional melting in **a**.

participation in the decompressional melting of asthenosphere. We therefore only modeled the melting of the DMM source to study the first-order compositional variations.

Our modeling shows that to produce the compositional ranges (Figs. 2, 3, Supplementary Fig. 4) and degrees of partial melting indicated by the majority of the Kangjinla peridotites (F = ~8.1–20.1%), the decompression melting must occur over a pressure range of at least 5 kbar (~15 km in depth; Fig. 4a). However, in the Kangjinla mantle profile these compositional ranges are expressed over a maximum depth of ~2 km (Fig. 4b). That is, the so-called "melt-depletion" gradient (~6%/km) of the mantle profile represents the minimum estimate, which is still significantly higher than those of modeled residual mantle columns (0.44–0.22%/km) at Tp of 1300–1450 °C (Supplementary Dataset 11). This means that the compositional variations of the Kangjinla mantle cannot be generated directly by adiabatic decompressional melting and lateral transposition from residual mantle columns as proposed by Plank and Langmuir[4].

In theory, an alternative scenario of proportional mechanical stretching and thinning of the residual mantle columns by at least 7–8 times may produce the observed compositional gradients of the Kangjinla mantle section. However, it is impossible for a lithospheric mantle section to be reduced in thickness by factors of 7–8 under an oceanic spreading center, considering the lack of reasonable dynamic forces[7] and constraints from thermal evolution models on the thickening oceanic lithosphere[9,10,42,43]. We also have not observed any structural evidence to suggest strong N-S compression or E-W stretching in the Kangjinla ophiolite (Fig. 1, Supplementary Fig. 1). In addition, even considering the contribution of ancient ultra-depleted components from the upwelling asthenosphere into the lithospheric mantle, it is still difficult to explain the gradual depletion features observed in this study (Figs. 2–4); meanwhile, Os-isotope investigations suggest that the ancient melting residues in the Luobusa mantle are heterogeneously distributed[35,36]. We therefore suggest that the vertical depletion of the Kangjinla ophiolitic mantle and probably of other ophiolites may be generated by additional processes, rather than simple adiabatic decompressional melting of asthenosphere or later proportional compression/extension of lithospheric mantle.

## Melt focusing and melt-peridotite interaction in asthenospheric upwelling column

Previous investigations of present-day mid-ocean-ridge samples and ophiolites both suggest that the melts extracted from a broad source mantle will migrate upwards and converge towards the narrow sub-axis zone[7,57,58], and will react extensively with the surrounding mantle during melt migration via diffusive and/or channelized flow[6,14,25,47,59,60]. Two main scenarios exist.

The first is that in the lithospheric mantle, mainly silica-saturated melts react with the peridotite resulting in cryptic metasomatic enrichments[14,18] and the addition of pyroxene/plagioclase[61], while minor silica-undersaturated melts react locally with the peridotite to form dunite lenses (via the reaction of pyroxene + silica-poor melt → olivine + silica-rich melt), which usually cut the foliation of the harzburgitic mantle[25]. In addition, observations from abyssal peridotites imply that the addition of olivine might occur pervasively in the oceanic lithospheric mantle[14,46], but it cannot explain the upward depletion of oceanic uppermost mantle.

The other scenario is that in the upwelling asthenospheric mantle, the dominative silica-undersaturated melts derived from deeper sources (down to garnet facies) migrate upwards and focus towards the center of an asthenospheric diapir, leading to the consumption of pyroxene, the addition of olivine and the formation of olivine-rich lenses conformable with the deformation patterns of the surrounding mantle[6,18,25,48].

To test if these two scenarios can generate the vertical gradual depletion and local anomalies observed from the Kangjinla ophiolitic mantle, we modeled the open-system reactions of two types of mafic melts with the most fertile lherzolite (KJL14-05A, with the highest Cpx mode and pyroxene/olivine modal ratio) from the SLZ, using the alphaMELTS program[52]. Sample KJL14-05A shows petrological features and major- and trace-element compositions of whole rocks and pyroxenes very similar to those of the Zedang lherzolites[24] and the Oman Type-I lherzolites[62], with a melting-residue origin weakly overprinted by cryptic metasomatic enrichments in some highly incompatible elements. The reactant compositions, conditions, and other parameters have been described in the Methods and Supplementary Dataset 10, while the modeled product compositions are shown in Supplementary Dataset 12.

Our modeling shows that the whole-rock compositional variations (including major oxides and fluid-unaffected trace elements) of the Kangjinla peridotites can be well reproduced by melt-peridotite reactions at 1300 °C in the shallow asthenosphere under a spreading center, but cannot form at lower mantle temperatures (e.g., 1000 °C and 1100 °C in the lithosphere and 1200 °C in the asthenosphere-lithosphere transition zone[43]; Fig. 5). The low-temperature melt-peridotite reactions mimic the trends of compositional enrichments mainly occurring in the lithospheric mantle (e.g., the decrease in MgO and increases in CaO, Ti, and Yb with progressive melt addition; Fig. 5). More importantly, the reactions at ~1300 °C with higher melt/rock ratios can result in stronger depletion in major oxides and intermediately-slightly incompatible trace elements until finally dunites are produced, showing gradually higher whole-rock MgO and FeO$_T$ as well as lower CaO, Ti, and Yb, as observed in the Kangjinla peridotites from the SLZ to the NDZ (Fig. 5). The abnormally-depleted peridotites affected by dunitization in the SLZ and the basal CHZ also can be formed by reactions at higher melt/rock ratios similar to those for the NDZ. In contrast, the more incompatible trace elements (e.g., LREE) display only mild enrichments because they are buffered by continuous addition of melts in the modeling (Supplementary Dataset 12), consistent with the LREE patterns of the studied peridotites (Supplementary Fig. 6).

We therefore propose that within the shallow upwelling asthenospheric column under an oceanic spreading center (e.g., ~4–8 kbar and ~1300 °C), the melts extracted from the deeper source mantle will flow and focus towards the sub-axis zone of rising residual mantle (represented by the sample KJL14-05A in the Kangjinla case; Fig. 6a). Larger amounts of melts will converge into the middle of the upwelling asthenospheric column to react at higher melt/rock ratios, while progressively less melt will react with the bilateral distal regions of the asthenospheric column (Fig. 6b). The melt-rock-ratio controlled compositional variations of the asthenospheric upwelling column at a given P-T condition are laterally symmetrical, with the middle part more depleted and the distal parts less depleted. In addition, the depletion anomalies represented by the dunitization-affected peridotites in the Kangjinla CHZ and SLZ can be explained by local melt accumulation and reaction in the bilateral distal regions of the asthenospheric column.

This melt-focused and compositionally-symmetrical column, as revealed by the Kangjinla case, can be several kilometers wide, consistent with those observed by geophysical studies of current mid-ocean ridges[57,58]. When this column rises it splits into two symmetrical parts, which will dynamically rotate ~90° in the mantle-flow regime to become the juvenile uppermost lithospheric mantle[3,7], with the primary middle part of the column at the top and the distal part toward the bottom (Fig. 6c). This perpendicular rotation of the split asthenospheric column can thus explain the vertical gradient in depletion and the local depleted anomalies in oceanic uppermost lithospheric mantle and ophiolitic mantle.

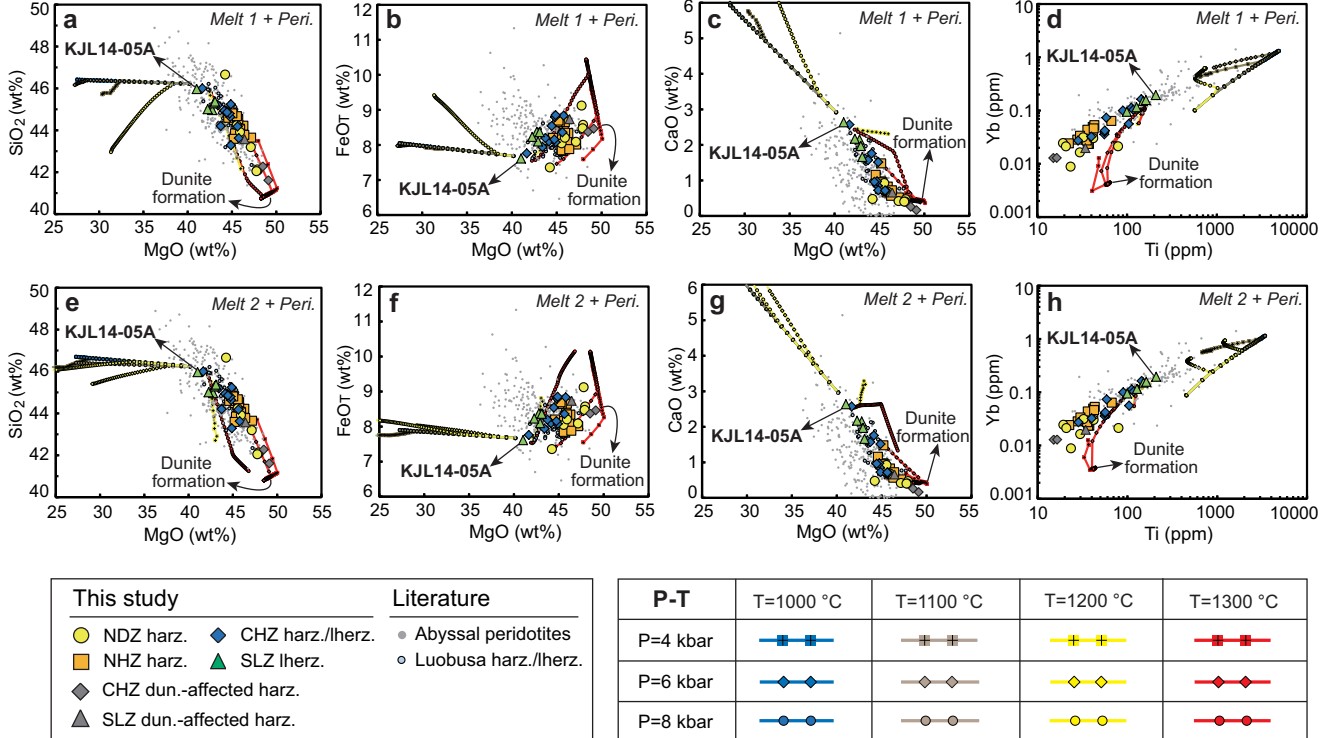

**Fig. 5 | Whole-rock compositional variations of the Kangjinla ophiolitic peridotites and melt-peridotite reaction modeling.** Variations of whole-rock MgO (wt%) versus SiO$_2$ (wt%, **a**, **e**), FeO$_T$ (wt%, **b**, **f**), and CaO (wt%, **c**, **g**) as well as whole-rock Ti (ppm) versus Yb (ppm, **d**, **h**) for the Kangjinla ophiolitic peridotites, compared to the solid-rock products modeled by open-system melt-peridotite reactions. The melt-peridotite reaction modeling was done using the alphaMELTS program[52]. Two scenarios of reaction are shown as "Melt 1 + Peridotite" in (**a**–**d**) and "Melt 2 + Peridotite" in (**e**–**h**). The modeling conditions (P and T) and the compositions of Melt 1, Melt 2, and Peridotite (lherzolite KJL14-05A) have been shown in Supplementary Dataset 10. Along with the addition of melts, the reactions at 1300 °C will continue until the formation of pure dunite. Gray and light-blue circles represent abyssal peridotites[13–15] and the published Luobusa peridotites[20, 32, 35, 38], respectively.

## Implications

Global dredged and drilled abyssal peridotites have been collected only from the epidermis of oceanic lithospheric mantle exposed along faults in mid-ocean ridges and transform systems[12–15], yet less than ~200 m deep into the mantle[16]. This shallowest oceanic mantle largely records the mantle source heterogeneity and the general complexity in partial melting and melt-mantle interaction[3–6,11–15,22,46–48], without vertical spatial constraints on a large scale. The short mantle stratigraphy (≤200 m in depth) revealed by drilled abyssal peridotites shows abundant gabbro, troctolite and pyroxenite dykes and rare dunite lenses within the harzburgitic mantle, mainly reflecting metasomatic enrichments by mafic melts at decreasing temperatures[13,16,61,63,64], consistent with the enrichment trends modeled by melt-peridotite reactions at temperatures lower than 1200 °C (Fig. 5).

This study of the Kangjinla ophiolitic profile (~2 km deep into the mantle) provides a deeper view of oceanic lithospheric mantle complementary to abyssal peridotites, and demonstrates that melt focusing and melt-mantle interaction at adiabatic asthenospheric conditions can produce lateral compositional variations in the asthenospheric upwelling column, which can subsequently rotate to become the upwards-depleted uppermost lithospheric mantle (Fig. 6). This model well explains the vertical gradual depletion and local anomalies in the several-km-deep mantle stratigraphy of classic ophiolites[17,18,21,44].

In addition, we provide a reconcilable solution for the debate regarding melt extraction and interaction modes in the mantle under oceanic spreading centers, i.e., fractal-tree dunite channel model[25] versus pervasive melt migration model[14]. The asthenospheric upwelling column beneath the axial zone of the spreading center can be regarded as a giant melt-focusing channel made up

of reactive lithologies transitional from lherzolite through harzburgite to dunite[48], with local melt accumulation regions forming so-called "dunite lenses"[25]. These form a multidimensional and heterogeneous melt-extraction system in the upwelling asthenosphere, with a first-order channel several km wide, which locally contains small-scale dunite tubes connecting as a high-porosity fractal-tree network. This understanding provides insights into the melt-extraction processes and the generation of oceanic lithospheric mantle beneath spreading centers with high melt fluxes (e.g., fast-spreading mid-ocean ridges and forearc/backarc centers), which are fundamental to and sustain the modern-style plate tectonics on Earth.

## Methods

### Whole-rock major- and trace-element analyses

Whole-rock major- and trace-element compositions were measured in the State Key Laboratory of Geological Processes and Mineral Resources (SKL-GPMR), China University of Geosciences (CUG, Wuhan).

Before the major-element analysis, each rock powder of 0.7 g was fully mixed with 5 g of flux (Li$_2$B$_4$O$_7$:LiBO$_2$ = 12:22), 0.3 g NH$_4$NO$_3$, 0.4 g LiF, and a few drops of LiBr. The mixed samples were melted at ~1050 °C and then quickly cooled as glass disks. After measuring these disks using a Shimadzu 1800 X-ray fluorescence spectrometer, raw data were processed using calibration curves produced by bivariate regression of data from 39 reference materials covering a wide range of silicate compositions. The measurement procedure and data quality were monitored by repeated analyses of two Chinese National ultramafic standards GBW07101 and GBW07102 (Supplementary Dataset 2). Unknown duplicates were measured to check the measurement reproducibility. Loss on ignition (LOI) was additionally measured. The

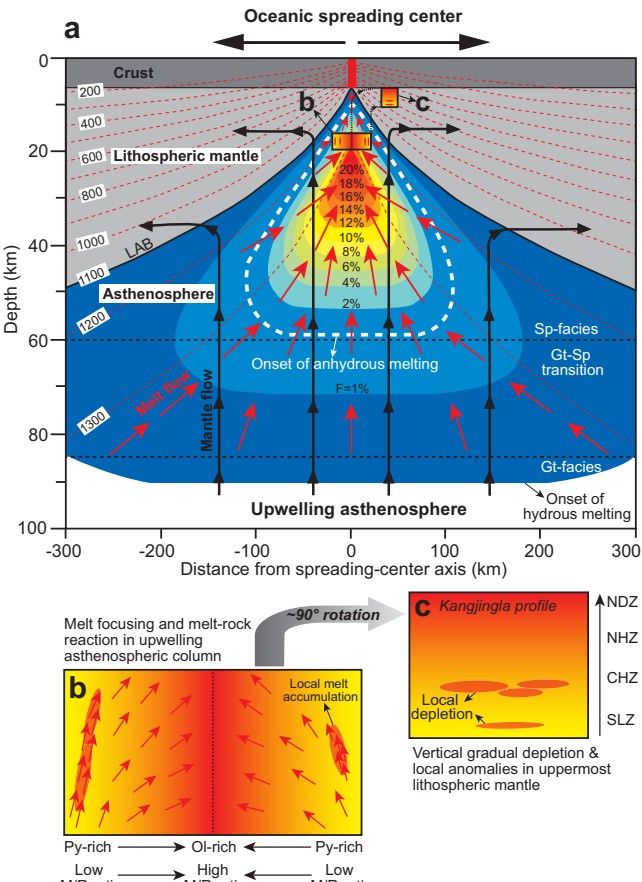

**Fig. 6 | Schematic cartoons (a-c) illustrate melt focusing and melt-peridotite interaction in the upwelling asthenospheric column under a typical oceanic spreading center.** Panel **a** shows a vertical cut plane (100 km deep and 600 km wide) of a sub-spreading-center regime far from transform faults, modified from Ligi et al.[7]. Lithosphere spreading and asthenosphere upwelling produce the color-coded decompressional melting regions with melting degrees (F) of 1–20% marked (**a**). Red dashed curves with temperatures from 1300 °C to 200 °C show the thermal structure of the lithosphere and asthenosphere[7,9,10,42], and the 1100 °C curve marks the lithosphere-asthenosphere boundary (LAB) as proposed by Niu and Green[43]. Two black dashed lines show the boundaries between spinel- and garnet-facies mantle and the garnet-spinel transitional zone (85–60 km) in between. The white dashed curve encloses the upwelling asthenospheric mantle where anhydrous melting occurs. The red short arrows show that the melts converge and focus into the sub-axis narrow zone under the spreading center. The black thick curves with black arrows display the flow patterns of upwelling asthenosphere. The residual mantle columns represented by the color-coded zones have the compositional gradients too small to be consistent with the Kangjinla situation (Figs. 2–4). A model of melt focusing and melt-rock reaction (pyroxene consumption and olivine addition) in the top region of the upwelling asthenospheric column (**b**) can well explain the vertical depletion and local depleted anomalies displayed in (**c**). The variations of melt/rock (M/R) ratios in the column result in the lithological and compositional variations, which form the laterally symmetrical mantle column (**b**). After it splits into two parts which each rotates ~90° to become the oceanic uppermost lithospheric mantle, the primarily more-depleted axial region forms the top of the lithospheric mantle section while the more-fertile bilateral region becomes the bottom (**c**), as observed in the Kangjinla and other ophiolites. Sp spinel, Gt garnet, Py pyroxene, Ol olivine, NDZ northern dunite zone, NHZ northern harzburgite zone, CHZ central harzburgite zone, SLZ southern lherzolite zone.

analytical relative standard deviations for the monitoring standards are less than 3%.

For the whole-rock trace-element analysis, ~50 mg rock powder of each sample was precisely weighed together with international standards (AGV-2, BHVO-2, BCR-2, GSP-2, and RGM-2). These samples were

fully digested by HF + HNO$_3$ in Teflon bombs, which were sealed by stainless-steel containers and heated at 190 °C for 48 h. The processed samples were dried and then added by 1.0 ml HNO$_3$, 1.0 ml ultra-pure water, and 1.0 ml indium solution until all the residues were dissolved. After dryness and cooling, each sample was diluted by a factor of 2000 using 2% HNO$_3$ before measurements on an Agilent 7700x inductively coupled plasma mass spectrometry (ICP-MS). The analytical accuracies for trace elements are better than 10%, monitored by the analyzed international standards (Supplementary Dataset 3).

### Mineral major-element analyses
Before the mineral major-element analyses, thick (~200 μm) sections of the Kangjinla peridotites were examined and imaged using a Nikon microscope and a Zeiss Sigma 300 field emission scanning electron microscopy (SEM). Back-scattered electron (BSE) images were taken by SEM, using a beam current of 20 nA, an accelerating voltage of 15 kV, and a beam size of ~1 μm. Mineral major-element compositions were determined using two electron microprobe analyzers (EMPA). The first is a JEOL JXA-8100 EMPA equipped with four wavelength-dispersive spectrometers at the Key Laboratory of Submarine Geosciences, Second Institute of Oceanography (MNR, China), and the second is a JEOL JXA-8230 EMPA with five wavelength-dispersive spectrometers at the SKL-GPMR of CUG (Wuhan). Both instruments used an accelerating voltage of 15 kV, a beam current of 20 nA, and a beam size of <1 μm. The peak counting time was 10 s for Na, Mg, Al, Si, K, Ca, Fe, and Cr, and was 20 s for Mn, Ti, V, and Zn. The background was counted for half as long as the peak, on both high- and low-energy background positions. The following standards were used: jadeite (Na), olivine (Si), diopside (Ca, Mg), sanidine (K), rutile (Ti), almandine garnet (Fe, Al), rhodonite (Mn), chromium oxide (Cr), and native metals V and Zn (V, Zn). The ZAF correction was used to calibrate the peaks by measurements of the above standards. The relative standard deviations of analyses on standards are less than 1%. During the analyses, the exsolved phases in pyroxene were avoided by careful selection of analytical spots.

### Mineral trace-element analyses
Trace-element compositions of clinopyroxene and orthopyroxene from the Kangjinla peridotites were measured using a laser ablation (LA)-ICPMS in the SKL-GPMR of CUG (Wuhan). A 193 nm RESOlution laser ablation system was attached to a Thermo iCAP-Q ICPMS for the analysis. For clinopyroxene, laser-ablation conditions of beam size 50 μm, pulse rate of 8 Hz, and an energy density of 3 J/cm$^2$ were applied. Each analysis includes 30 s on background at the beginning and then 40 s collection of sample signals. Multiple reference materials (NIST 610, NIST 612, BIR-1G, BCR-2G, and BHVO-2G; measured and recommended values are shown in Supplementary Datasets 8, 9) were used as external standards without the application of an internal standard for data reduction[65]. Selection and integration of background and sample signals, the calibration of fractionation derived from ablation, transportation and excitation processes, and the matrix effect on the data were all processed using the off-line program ICPMSDataCal 11[65]. The data collection and calibration processes for orthopyroxene were similar to those for clinopyroxene, except for the usage of a laser beam size of 130 μm. The analytical uncertainty is better than 5% for REEs and 10% for the remaining elements (1 s level).

### Thermodynamic modeling
Isentropic decompressional fractional melting of the DMM source[49] was modeled using the pMELTS version[51] of the alphaMELTS 1.9 software package[52], at mantle potential temperatures (Tp) of 1300 °C, 1350 °C, 1400 °C, and 1450 °C[50] (parameters and results are shown in Supplementary Datasets 10 and 11, respectively). The step length of the decompression is set to 0.1 kbar, and the threshold of melt fraction for melt extraction at all steps is set to 0.005. The instantaneous melts formed at each step stay in thermo-chemical equilibration with the

corresponding solid, and then the melt portions beyond the threshold will be extracted and pooled step by step to form the integrated melts, which are usually out of equilibrium with the melting residues.

Open-system reactions of two types of mafic melts with the lherzolite KJL14-05A from the Kangjinla SLZ have been modeled using the alphaMELTS program[52]. Melt 1 is represented by the integrated melts extracted from the DMM source with a Tp of 1350 °C where the melting starts at the crosscut of solidus with the mantle adiabat ($P = 21.0$ kbar) and ends at the depth of spinel exhaustion ($P = 8.3$ kbar). We consider a two-dimension scenario where the melting regime has a trapezoidal shape and the composition of the integrated melts from such a melting regime can be calculated via weighted integration of the instantaneous melts[66]. The step length of decompression and threshold for melt extraction are as described before. The major-element compositions of each extracted melt increment are determined by the phase equilibrium at the corresponding decompression step, and the concentrations of selected trace elements are calculated from the phase assemblages and the compiled mineral-melt partition coefficients (Supplementary Dataset 10). The temperature of the integrated melts is set to be the same as that of the last melt increment at the melting adiabat (~1320.23 °C). Melt 2 represents the integrated melts similar to Melt 1 except for ending at the pressure of 5.3 kbar when all Cpx is consumed and a melt temperature of 1295.98 °C (Supplementary Dataset 10). The pressures (4, 6, and 8 kbar) and temperatures (1000, 1100, 1200, and 1300 °C) of peridotitic mantle have been set up to mimic the conditions of a very short mantle column (a few kilometers in length, similar to the Kangjinla mantle profile) typically in the uppermost lithospheric mantle and the top of the upwelling asthenospheric mantle under spreading centers[7,9,10,42].

At each mantle P-T combination, melts with each parcel of 4 g are added iteratively to the lherzolite KJL14-05A with an initial mass of 100 g. After the melt-solid bulk system attains thermodynamic equilibration, the producing melts will keep in the bulk system if the melt fraction is below 0.005. Otherwise, the melt portion beyond this threshold will be extracted from the system. The resultant system with ≤0.005 melt fraction is then augmented by a new parcel of 4 g melt, and undergoes a new round of melt-rock equilibration and possible melt extraction. This iteration will stop when the total melt addition reaches 200 g, or until the open-system reaction ceases due to energy limits. The modeled whole-rock solid compositions are shown in Supplementary Dataset 12.

## Reporting summary

Further information on research design is available in the Nature Portfolio Reporting Summary linked to this article.

## Data availability

The authors declare that the data generated in this study are provided in Supplementary Datasets 1–12.

## Code availability

The modeling codes are included in the Source Code 1.

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

## Acknowledgements

We thank X. Zhou, Z.Y. Li, W.W. Wu, H. Liang, A.B. Lin, J.X. Huang, S.Y. Cao, W. Chen, T. Luo, and B. Xia for assistance during field work, geochemical analyses, and figure drafting. This study was supported by the Second Tibetan Plateau Scientific Expedition and Research Program (STEP) (2019QZKK0702 to J.P.Z. and Q.X.), the National Natural Science Foundation of China (42073030 and 41873032 to Q.X.), the MOST Special Fund from the State Key Laboratory of Geological Processes and Mineral Resources, China University of Geosciences (MSFGPMR2022-6 to Q.X. and H.K.D.), and the Australian Research Council (ARC) Centre of Excellence CCFS. This is contribution 1746 from the ARC Centre of Excellence for Core to Crust Fluid Systems (www.ccfs.mq.edu.au) and 1515 from the GEMOC Key Centre (www.gemoc.mq.edu.au) and is relevant to IGCP Project 662.

## Author contributions

Q.X., J.P.Z., W.L.G., and S.Y.O'R. co-designed this project. Q.X., H.D.Z., and L.W. collected the samples and carried out the geochemical analyses. H.K.D. conducted the thermodynamic modeling. Q.X. wrote the manuscript with contributions from H.K.D., J.P.Z., W.L.G., and S.Y.O'R.

## Competing interests

The authors declare no competing interests.
