## [Peer Review File · Nature Communications]

Vertical depletion of ophiolitic mantle reflects melt focusing and interaction in sub-spreading-center asthenosphereReviewer #1 (Remarks to the Author):

Dear Editor,

This letter includes my review to the manuscript "Vertical depletion of ophiolitic mantle decodes melt focusing and interaction in the asthenospheric column under oceanic spreading centers" by Qing Xiong and coauthors. The comments are detailed in the following pages (attached as separate pdf file), and can be followed in the annotated pdf file (also attached), where you can find additional minor corrections and some more explanations.

The authors present an extremely high-quality data set of a long mantle section exposed in the Loubusa ophiolite, South Tibet. They use geochemical data, mainly whole rock major element compositions and trace element compositions of mineral phases, to discuss the origin of a Km-scale stratification in this mantle section. The rationale of the project is clear, and the authors present a model that, if validated, can be of interest for a large audience, potentially suitable for Nature Communications. The text is well written, the abstract explains well the idea and the discussion are clearly presented, although I suggest adjusting the data presentation that should avoid unnecessary details and crude numbers. However, despite the high quality and the importance of the data set, I regret to not be able to recommend publication in the present form. The model they propose is new, but not fully sustained by the data (see comments 2 and 4) and, more importantly, in my knowledge cannot be used to "reveal the dynamic processes responsible for the compositional features of the uppermost lithospheric mantle under oceanic spreading centers", simply because it does not recall the composition of the uppermost ocean mantle exposed at present day MOR (comments 1 and 4). Nonetheless, the study can be revised to make it suitable to the journal. In detail, I suggest to i) better constrain the geochemical effect of the melt-rock reaction model, reproducing the trace element compositions of pyroxene; and ii) reconsider the general message in the framework of the present data on the architecture of the abyssal mantle, bringing new fundamental insights to the dichotomy between ophiolitic and MOR stratigraphy.

- What are the noteworthy results?

The authors reconstruct a kilometers-thick section of upper mantle having an exceptionally consistent geochemical stratification referable to depletion or melt-rock reaction processes.

- Will the work be of significance to the field and related fields? How does it compare to the established literature? If the work is not original, please provide relevant references.

Yes, if the model would also explain the trace element signatures of the pyroxene. In this regard, the model is original and new.

- Does the work support the conclusions and claims, or is additional evidence needed? The conclusions are not entirely supported by the geochemical model (see below), the claims need to be reconsidered in the light of present-day MOR mantle

- Are there any flaws in the data analysis, interpretation and conclusions? Do these prohibit publication or require revision?

In the present form, the conclusion and the implication of the work preclude publication on Nature Communications

- Is the methodology sound? Does the work meet the expected standards in your field? Definitely yes!! The sample strategy is clear, the analytical work is first-class. The data treatment is correct.

- Is there enough detail provided in the methods for the work to be reproduced?

Totally yes. I cannot judge the extent by which the samples represent the field exposure of the rocks, based on the literature, there are other lithologies and rocktypes that deserve attention (boninite-like magmas, chromitites). This requires a reviewer expert in the regional geology.

As such, the manuscript does not presently meet the requirements for the journal, but

has high potential to be published after some important modifications. I will be happy to re-evaluate and recommend a second version of this study, if requested by the editors.

I hope the authors may find my comment of utility to increase the quality and the importance of their unbelievable effort.

Best regards,

Alessio Sanfilippo

Associate Professor in Petrology,
Department of Earth and Environmental Science,
University of Pavia,
Via Ferrata, 1, 27100 Pavia, Italy

In their review of the first version of this manuscript, reviewer #1 added some comments to the manuscript file. These comments were forwarded to the authors, who replied as included in this Peer Review File

Reviewer #2 (Remarks to the Author):

Review of Vertical depletion of ophiolitic mantle decodes melt focusing and interaction in the asthenospheric column under oceanic spreading centers

By Qing Xiong, Hong-Kun Dai, Jian-Ping Zheng, William L. Griffin, Hong-Da Zheng, Li Wang, and Suzanne Y. O'Reilly

The authors present new major and trace element results for peridotites from the Kangjinla massif (Luobusa ophiolite), with the specificity of having applied a cross-section sampling strategy from the deepest to shallowest levels of the mantle section (about 2km vertically). A vertical petrological and geochemical evolution is described, with more refractory characters upward, defined by the decrease in the clinopyroxene content and more and more depleted chemical compositions. Based on this observation, the authors performed thermodynamical models to show that this compositional gradient cannot directly result from decompressional melting, but requires melt/rock interaction in the asthenosphere prior to mantle flowing and lithospherization. The data are of good quality and the methods are well detailed. The manuscript and figures are very clear and easily understandable. The proposed model is interesting and elegant to account for the observed spatial chemical variation. However, a few points need to be addressed, discarded or discussed, before making this manuscript suitable for publication in Nature Communications. I detail here below these few points as questions or comments.

- Origin of clinopyroxene: residual origin or refertilization?

Clinopyroxene-bearing harzburgites or lherzolites have also been observed in other ophiolites in the deepest levels of mantle sections, close to the thrust contact, such as in the Oman (Godard et al., 2000; Takazawa et al., 2003; Prigent et al., 2018) or Bay of Island (Girardeau and Nicolas, 1981) ophiolites. In some cases clinopyroxene (cpx) is better explained as a result of melt-peridotite reaction, possibly 'at near-solidus conditions along the lithosphere-asthenosphere boundary' (in the abstract of Godard et al., 2000), than as a residual mineral phase.

In your study, you likely observe cpx-bearing peridotites in the deepest SHZ unit. You consider this cpx as a residual phase, making the host rocks your more 'fertile' samples, but you do not really discuss the possibility of a reactive origin. It should be done, at

least in one paragraph in the discussion, as it is a critical point to support your following interpretation - 'the most fertile harzburgite (KJL14-05A)' (line 273) is used for modelling. In the same way, please better describe the texture of cpx in the results section as it is now not detailed excepting 'sometimes occurring as porphyroblasts' (Lines 109-110) (associated with Opx?). A good argument supporting the residual origin, and you mention it several times, is that the cpx disappears upward gradually, i.e. it is not restricted to the vicinity of the few hundred meters at the base of SHZ close to Langjiexue Group (Fig 1), but it should be stated if it is your main argument to favour the residual vs. refertilization origin.

This is the main criticism I have as if the residual origin of cpx is not clearly proven, and finally is a melt-rock reaction/refertilization product, all the following discussion-interpretation fall.

- Differences between your cross-sections?

In Figure 1 we can see that you sampled two (sub-)parallel cross-sections across the NHZ and CHZ units. However, there is no distinction in the chemical plots. A slight weakness of your work is that it is based on a solely synthetic cross-section, so we cannot really envision if the global depletion upsection you describe applies to the whole ophiolite or not, while several cross-sections could have shown that. A way to partially overcome this could be to make the distinction between the different cross-sections in your plots, to show if the data from respective cross-section complement each other to form a common trend, or if it is more variable/complicated at small-scale.

- High-density sampling, and continuity between units

Another point is that you did not sample the base of the CHZ just above the SHZ - there is no direct continuity in the sampling from the cpx-rich SHZ to the cpx-poor or cpx-free CHZ while there is a clear, distinct chemical shift between these two units (Fig. 2).

Accordingly, there is a gap of a few hundred meters between the top of SHZ near 1750 m and the bottom of CHZ near 1350 m (Fig. 2 and Suppl Table 1). Are there any chemical data from literature that could fill this gap and expand/confirm the general trend you describe? Surprisingly you compare your data with compositions of abyssal peridotites, ok with that, but not with compositions of previous works on the same Luobusa/Kangjinla ophiolite.

An additional observation is that you report dunites at the base of the CHZ (Suppl Fig. 1). It is also commonly observed, together with the presence of cpx-bearing harzburgites, just above the basal thrust contact of ophiolites (e.g. in Oman, Klaessens et al., 2021). These basal dunites are usually interpreted as formed during the intra-oceanic thrusting leading to the ophiolite's obduction. If so, how to be sure about the continuity from the SHZ and CHZ, especially regarding the lack of samples and data on 400 m in thickness at the base of CHZ? Could the thrust fault have developed just below the CHZ in the eastern part of the massif? (i.e. just to the south of your blue samples in Fig. 1c).

- Tilt of the Kangjinla ophiolitic units?

Lines 83-84: 'This zonation reveals that the mantle section was overturned during its emplacement'.

It seems there is a petrological logic in the successive units you describe. However, a critical point not addressed in your manuscript is if the cross-section is really vertical, perpendicular to the paleo-Moho/paleo contacts, or not. What I mean is that if there is an angle between your sampling line and the paleo contacts, it could allow to under/over-estimate the observed 'vertical' depletion (12% as indicated in the abstract Lines 22-23), and accordingly bias the melting pressure range (5 kbars) and thus the depth/thickness (15 km) on which partial melting potentially occurred (Line 239) and estimated from your thermodynamical modelling. You finally conclude that it is unlikely that the data reflect partial melting event as you favour the melt/peridotite processes, but such estimation in the depletion % and/or pressure could be biased. How to be sure that your sampling overcome or took into account a possible structural tilt of the units?, and thus how to be sure your sampling represent 'a high-resolution view of the lithological and compositional variations in a section of the oceanic lithospheric mantle'

(Lines 58-59)?

- Consistency between the model and geological observations?

Lines 314-310: 'When this column rises it splits into two symmetrical parts, which will dynamically rotate $\sim 90^\circ$ in the mantle-flow regime to become the juvenile uppermost lithospheric mantle^{3,7}, with the primary middle part of the column at the top and the distal part to the bottom (Fig. 6c). This perpendicular rotation of the split asthenospheric column can thus explain the vertical gradient in depletion and the local depleted anomalies in oceanic uppermost lithospheric mantle and ophiolitic mantle.' This is an elegant model. At the same time, it has been observed both in present-day oceans along oceanic spreading centres (e.g. Dick and Natland, 1996; Godard et al.; 2008) and in un-transposed ophiolitic massifs (e.g. Maqsad area in Oman, Rabinowicz et al., 1987; Abily and Ceuleneer, 2013; Rospabé et al., 2017) that the transition between the mantle section and the base of the crust is extensively made of dunites. This is what you describe for the NHZ, so it's tempting to interpret this unit as the original (paleo-)Moho. How to account this observation with the idea that the deepest mantle section has been rotated by 90° ? A decoupling between the NHZ and other units, while the gradational chemical evolution over all units is observed?

- Additional comments

Line 26: 'melt/rock variations'.

'melt/rock ratio variations' would be clearer.

Lines 49-52: 'For example, the fractal dunite melt-channel system in the upwelling residual mantle has been revealed by studies of ophiolites (mainly the Oman example) to illustrate the melt-extraction processes and mantle dynamics under oceanic spreading centers^{6,24}'.

Has been 'revealed' is not exactly the proper term. The fractal channel has not been directly observed in ophiolites, especially Oman, but was conceptualised from the observation of horizontal channels possibly transposed off-axis, which are much more common in ophiolitic sequences than vertical channels - see the Figure 16 caption in Braun and Kelemen 2002 (the black box in the top right corner that locates what was really observed on the field). The fractal concept has actually been more supported by numerical modelling (theory) than by strong field observations (practice). It is not a big point here, but maybe simply replace 'revealed' by 'conceptualised', 'theorised', 'proposed', or another word.

Lines 105-106: 'porphyroblastic textures and plastic deformation'

This short description supports mantle flowing related to off-axis transposition of asthenospheric mantle described in your model. However, we cannot really check that on the four pictures in Suppl Fig 1 as we do not see mineral preferential orientations or elongated minerals. Could it be possible to add additional pictures? Are there different deformation features between samples from NHZ on one hand, and other samples on the other hand? (see my comment about NHZ above).

Lines 326-332: 'However, within the same sampling site on a spreading center, such as the narrow $\sim 22-24$ Ma sector in the Vema Fracture Zone of the Mid-Atlantic Ridge, spinel Cr# values of the abyssal peridotites also show a large range, indicating major compositional variations⁵⁹. The large variations are difficult to explain by the almost consistent physiochemical conditions and sources within this small domain, but may be easily resolved by the sampling of a short profile of the vertical uppermost lithospheric mantle produced by the processes proposed in this study (Fig. 6).'

El Dien et al. (2019) recently shown that the chemistry of Cr-spinel could possibly reflect metasomatism only, and not to be a mantle melting indicator.

The methods are well presented and detailed. Values for a few standards are given in Suppl Table 2 following the whole rock major element compositions. You also mention in the text having analysed multiple standards during laser ablation analyses (Lines 385-386). Mean values of repeated analyses of these standards could be given in Suppl Table

7 or 8 after clinopyroxene or orthopyroxene trace element data.

**I hope my comments/questions will help you to clarify some points of your manuscript.
Best regards**

References

- Abily, B., & Ceuleneer, G. (2013). The dunitic mantle-crust transition zone in the Oman ophiolite: Residue of melt-rock interaction, cumulates from high-MgO melts, or both?. *Geology*, 41(1), 67-70.**
- Braun, M. G., & Kelemen, P. B. (2002). Dunite distribution in the Oman ophiolite: implications for melt flux through porous dunite conduits. *Geochemistry, Geophysics, Geosystems*, 3(11), 1-21.**
- Dick, H. J., & Natland, J. H. (1996). Late-stage melt evolution and transport in the shallow mantle beneath the East Pacific Rise. In *proceedings-ocean drilling program scientific results* (pp. 103-134). National Science Foundation.**
- Gamal El Dien, H., Arai, S., Doucet, L. S., Li, Z. X., Kil, Y., Fougereuse, D., ... & Hamdy, M. (2019). Cr-spinel records metasomatism not petrogenesis of mantle rocks. *Nature communications*, 10(1), 1-12.**
- Godard, M., Lagabrielle, Y., Alard, O., & Harvey, J. (2008). Geochemistry of the highly depleted peridotites drilled at ODP Sites 1272 and 1274 (Fifteen-Twenty Fracture Zone, Mid-Atlantic Ridge): Implications for mantle dynamics beneath a slow spreading ridge. *Earth and Planetary Science Letters*, 267(3-4), 410-425.**
- Godard, M., Joussetin, D., & Bodinier, J. L. (2000). Relationships between geochemistry and structure beneath a palaeo-spreading centre: a study of the mantle section in the Oman ophiolite. *Earth and Planetary Science Letters*, 180(1-2), 133-148.**
- Klaessens, D., Reisberg, L., Joussetin, D., Godard, M., & Aupart, C. (2021). Osmium isotope evidence for rapid melt migration towards the Moho in the Oman ophiolite. *Earth and Planetary Science Letters*, 572, 117111.**
- Prigent, C., Agard, P., Guillot, S., Godard, M., & Dubacq, B. (2018). Mantle wedge (de) formation during subduction infancy: Evidence from the base of the Semail ophiolitic mantle. *Journal of Petrology*, 59(11), 2061-2092.**
- Rabinowicz, M., Ceuleneer, G., & Nicolas, A. (1987). Melt segregation and flow in mantle diapirs below spreading centers: evidence from the Oman ophiolite. *Journal of Geophysical Research: Solid Earth*, 92(B5), 3475-3486.**
- Rospabé, M., Ceuleneer, G., Benoit, M., Abily, B., & Pinet, P. (2017). Origin of the dunitic mantle-crust transition zone in the Oman ophiolite: The interplay between percolating magmas and high-temperature hydrous fluids. *Geology*, 45(5), 471-474.**
- Takazawa, E., Okayasu, T., & Satoh, K. (2003). Geochemistry and origin of the basal lherzolites from the northern Oman ophiolite (northern Fizh block). *Geochemistry, Geophysics, Geosystems*, 4(2).**

Reviewer #3 (Remarks to the Author):

Dear Editors and Authors:

Thanks for giving me the chance to review the manuscript NCOMMS-22-07539 "Vertical depletion of ophiolitic mantle decodes melt focusing and interaction in the asthenospheric column under oceanic spreading centers".

I have read this manuscript with great interest. The authors have presented us a whole picture on the vertical compositional variations of oceanic mantle in a spreading center, showing upward depletion and local depleted anomalies. In general, this manuscript is well written, I would suggest acceptance with a minor revision for this manuscript. I only have some minor comments as listed below:

(1) Line 39-42: The authors suggest that oceanic crust can be studied through research on fossil ophiolites, but the lithospheric mantle is mainly studied from abyssal peridotites, which is contradictory. There are plenty of studies on oceanic mantle from

global ophiolitic peridotites.

(2) Line 63: Normally this section should go immediately after the "introduction", rather than as a part of the "result" section.

(3) Line 66-68: I would suggest the authors to put some examples of peridotites from the northern sub-belt of YZSZ to give the readers a whole picture on the YZSZ ophiolites, and put the related references here.

(4) Line 131-133: It seems unreasonable that serpentinization will not affect the composition of peridotites.

(5) Line 154, 158: Please verify the number of significant digits here.

(6) Line 292-295: It is confusing for me here. How can silica saturated melts reacting with mantle peridotites to produce the more depleted endmembers. It is not clear here what is the different roles played by these different types of magma in the generation of the chemical trend of the Kanjingla peridotites.

Best regards

Dongyang Lian

Nanjing University.

In their review of the first version of this manuscript, reviewer #3 added some comments to the manuscript file. These comments were forwarded to the authors, who replied as included in this Peer Review File

Revision Notes

Responses to comments from Reviewer #1 (Dr. Alessio Sanfilippo)

Reviewer #1 (Remarks to the Author):

Dear Editor,

This letter includes my review to the manuscript “Vertical depletion of ophiolitic mantle decodes melt focusing and interaction in the asthenospheric column under oceanic spreading centers” by Qing Xiong and coauthors. The comments are detailed in the following pages (attached as separate pdf file), and can be followed in the annotated pdf file (also attached), where you can find additional minor corrections and some more explanations.

The authors present an extremely high-quality data set of a long mantle section exposed in the Loubusa ophiolite, South Tibet. They use geochemical data, mainly whole rock major element compositions and trace element compositions of mineral phases, to discuss the origin of a Km-scale stratification in this mantle section. The rationale of the project is clear, and the authors present a model that, if validated, can be of interest for a large audience, potentially suitable for Nature Communications.

The text is well written, the abstract explains well the idea and the discussion are clearly presented, although I suggest adjusting the data presentation that should avoid unnecessary details and crude numbers.

Our response: We have significantly shortened the descriptions of data, and avoided unnecessary details and crude numbers in the revision (please see lines 136-166 in the revision).

However, despite the high quality and the importance of the data set, I regret to not be able to recommend publication in the present form. The model they propose is new, but not fully sustained by the data (see comments 2 and 4) and, more importantly, in my knowledge cannot be used to “reveal the dynamic processes responsible for the compositional features of the uppermost lithospheric mantle under oceanic spreading centers”, simply because it does not recall the composition of the uppermost ocean mantle exposed at present day MOR (comments 1 and 4). Nonetheless, the study can be revised to make it suitable to the journal.

Our response: Abyssal peridotites have been dredged and drilled from the exposed shallowest mantle in present-day mid-ocean ridges or along transform faults (e.g., Dick et al., 2003; Niu, 2004; Warren, 2016). The reported deepest oceanic drill hole into the mantle is only ~200 m in depth at the Mid-Atlantic Ridge (Cannat et al., 1995). The mantle stratigraphy (generally ≤ 200 m) revealed by the oceanic drilling shows abundant gabbro, troctolite and pyroxenite dykes and rare dunite lenses within the harzburgitic mantle, reflecting metasomatic enrichments by mafic melts at decreasing temperatures (Cannat et al., 1995; Dick & Natland, 1996; Tartarotti et al., 2002; Kelemen et al., 2004; Warren, 2016). These characteristics are consistent with the enrichment trends at temperatures lower than 1200 °C in our melt-peridotite modeling (**Figure 5** in the revision). However, they are clearly different from the upward compositional depletion shown by well-preserved ophiolites, with mantle depths of several to tens of kilometers, such as Kangjinla (this study), Oman (Godard et al., 2000), Troodos (Batanova & Sobolev, 2000) and Bay of Islands (Suhr & Robinson, 1994). Please also see our following responses to the comments (1), (2) and (4) from Reviewer#1. We believe that our revision has carefully addressed these comments and made it suitable for Nature Communications.

In detail, I suggest to i) better constrain the geochemical effect of the melt-rock reaction model, reproducing the trace element compositions of pyroxene; and ii) reconsider the general message in the framework of the present data on the architecture of the abyssal mantle, bringing new fundamental insights to the dichotomy between ophiolitic and MOR stratigraphy.

Our response: Thanks for the two constructive suggestions. 1) New modeling of open-system melt-peridotite reactions (including both major elements and selected trace elements) has been done (see lines 406-442 in the **Methods** of the revision). Two types of integrated melts with more reasonable compositions and temperatures, as well as three pressures and four temperatures of peridotitic mantle, have been considered to mimic the most probable situations of melt-peridotite reactions in the topmost asthenospheric column under spreading centers. In addition, because of the strong effect of sub-solidus compositional re-equilibration between olivine, orthopyroxene, clinopyroxene and spinel, we do not compare the measured and modeled clinopyroxenes directly, but compare their whole-rock compositions (see **Figure 5** in the revision).

2) The compositional structure of present-day abyssal mantle has been carefully compared with the ophiolitic stratigraphy, and we have discussed the so-called “dichotomy” between mid-ocean-ridge (MOR) and ophiolitic stratigraphy (see lines 318-334 in the revision). In addition, a reconcilable solution for the debate regarding melt extraction and interaction mechanisms in the oceanic mantle under spreading centers, i.e., fractal-tree dunite channel model (e.g., Kelemen et al., 1995) vs pervasive melt migration model (e.g., Niu, 2004), has been provided in the revision. We have proposed that the asthenospheric upwelling column beneath the axial zone of the spreading center can be regarded as a giant melt-focusing channel made up of reactive lithologies transitional from lherzolite through harzburgite to dunite (Dick & Zhou, 2015), with local melt accumulation regions forming so-called “dunite lenses” (Kelemen et al., 1995; **Figure 6** in the revision). A multidimensional and heterogeneous melt-extraction system can thus be established in the upwelling asthenosphere, with a first-order several-km-wide channel that locally contains small-scale dunite tubes connecting in a high-porosity fractal-tree network. We believe that the new perspectives in the revision will provide significant implications for the melt extraction processes and the formation mechanisms of oceanic uppermost mantle.

- What are the noteworthy results?

The authors reconstruct a kilometers-thick section of upper mantle having an exceptionally consistent geochemical stratification referable to depletion or melt-rock reaction processes.

- Will the work be of significance to the field and related fields? How does it compare to the established literature? If the work is not original, please provide relevant references.

Yes, if the model would also explain the trace element signatures of the pyroxene. In this regard, the model is original and new.

- Does the work support the conclusions and claims, or is additional evidence needed?

The conclusions are not entirely supported by the geochemical model (see below), the claims need to be reconsidered in the light of present-day MOR mantle

- Are there any flaws in the data analysis, interpretation and conclusions? Do these prohibit publication or require revision?

In the present form, the conclusion and the implication of the work preclude publication on Nature Communications

- Is the methodology sound? Does the work meet the expected standards in your field?

Definitely yes!! The sample strategy is clear, the analytical work is first-class. The data treatment is correct.

- Is there enough detail provided in the methods for the work to be reproduced?

Totally yes. I cannot judge the extent by which the samples represent the field exposure of the rocks, based on the literature, there are other lithologies and rock types that deserve attention (boninite-like magmas, chromitites). This requires a reviewer expert in the regional geology.

As such, the manuscript does not presently meet the requirements for the journal, but has high potential to be published after some important modifications. I will be happy to re-evaluate and recommend a second version of this study, if requested by the editors.

I hope the authors may find my comment of utility to increase the quality and the importance of their unbelievable effort.

Major comments:

(1) General message of the study and its applicability to MOR.

The message of this study, which would fully justify publication on Nature Communication, is that vertical melt focusing can produce a horizontal stratification of the residual mantle exposed at ocean ridges. The model recalls but surpasses the classical fractal trees model, where high permeability (dunites) channels serve as migration paths and are then transposed horizontally along the crust-mantle boundary. Here the authors suggest that this process works at much higher scale, and a giant melt focusing channel may form at the central portion of the melting region, then split and transposed laterally to form a km-scale stratigraphy of dunites, cpx-poor harzburgites and lherzolites. The idea is great, and new. However, if correct, we should find exclusively dunite and “relic enclaves of harzburgites” for the entire uppermost 200 m. And looking at the geological map, this dunitic horizon would be continuously distributed, at least at high melting degrees.

Our response: Thanks for these suggestions. We have described the field lithology and petrography more clearly in the revision (lines 130-134). In fact, the harzburgites do exist in the Kangjinla northern dunite zone (NDZ) as relict enclaves (**Figure 1** in the revision), which show diffuse boundaries toward the dunites. The NDZ harzburgites commonly have the most abundant olivine, the least modal orthopyroxene and rare clinopyroxene, compared to peridotites in other zones (**Supplementary Table 1**). The pyroxenes display resorbed shapes and smaller sizes, while the spinels form trails and become rounded or euhedral, similar to those in the typical dunites (**Supplementary Figure 1b-1d**). These petrographic features, together with those of other zones, demonstrate a gradual transition from lherzolites through harzburgites to dunites from the southern lherzolite zone (SLZ) to the NDZ in general.

The authors are right when they claim that the lack of systematic sampling of AP hampers the definition of an upper mantle stratigraphy of MOR. However, a large effort has been spent in the last decades to drill sections of mantle exposed at the seafloor. The deepest hole into the mantle is 200 m-thick (exactly like the NDZ in this study) at Site 920 in the Kane area, and several holes have been drilled at 15.20°N FZ (MAR). The mantle peridotites are very heterogeneous, and none of these holes reported large quantities of dunites, not even comparable to the NDZ or NHZ. Dunite is a relatively common lithology at Hess Deep Site 895, iconic of a crust-mantle boundary of fast spreading ridge stratigraphy. However, even here, dunites are included in harzburgites and in turn include troctolites and gabbros. Tectonic denudation could not have removed thick layer of mantle, and even at CMB the basement after few tens of meters seems to be mostly formed by residual harzburgites (see figure on the right). Highly refractory peridotites to dunites have been described at the 15.5°N FZ at MAR (Kelemen et al, 2004), where, again, they are associated to troctolites and gabbros, but the chemical depletion is considered to have predated the last melting event at MAR (Harvey et al., 2006; 2010).

Our response: Thanks for these constructive suggestions. The contents related to abyssal peridotites in lines 38-44 and in the **Implications** (lines 318-327) have been revised. Abyssal peridotites only

reveal the shallowest oceanic lithospheric mantle limited to the depth of ~200 m until now (Cannat et al., 1995), while well-preserved ophiolites can provide mantle stratigraphy deeper than several kilometers, such as the Kangjinla ophiolite in this study, as well as other representative ophiolites in Oman, Troodos and Bay of Islands. We agree that the shallowest mantle beneath oceanic crust (if exists) generally contains gabbros or troctolites, a typical feature of the crust-mantle transition zone both under present-day spreading centers (e.g., Cannat et al., 1995; Dick & Natland, 1996; Tartarotti et al., 2002; Kelemen et al., 2004) and in representative ophiolites (e.g., Kelemen et al., 1997a; Suhr et al., 1998). These lithological associations can be well reproduced by melt-peridotite reactions at temperatures decreasing to below 1200 °C (mainly within the lithosphere levels) in our modeling (**Figure 5** in the revision). This shallowest epidermis of oceanic lithospheric mantle should theoretically show lithological and compositional variations different from the several-km-deep ophiolitic mantle stratigraphy, which commonly exhibits upward depletion (e.g., this study; Suhr & Robinson, 1994; Batanova & Sobolev, 2000; Godard et al., 2000).

Systematic dredging of mantle peridotites has been carried out at the Vema lithospheric section (Brunelli et al., 2018) and Doldrums FZ (Sani et al., 2020), for an upper mantle section covering a 30 Ma-long time span. Again, dunites are extremely rare, and most peridotites show a purely residual character, with no obvious evidence of infiltrating melt. The original mineralogy of these highly serpentinized rocks is difficult to obtain, but high degrees of melt infiltration or olivine addition would have caused local variations in mineral (spl, pyr) chemistry as revealed in replacive dunites. Even more importantly, the residual character is heterogeneous at km-scale (see also Sani et al, 2020 for an examples), and locally, is coupled with heterogeneous Nd-Hf isotopes (Stracke et al., 2011) witness of old depletion events. Sani et al. (under review on Nat Geosc) will hopefully soon show that the melting degrees are primary related to an initial heterogeneity as revealed by radiogenic isotopes, sustaining a concept recently advanced by Urann et al. (2019) Nature Communications for highly residual peridotites from 16.5°N MAR. Hence, based on existing data, I am skeptical that the stratification proposed by the authors characterizes the oceanic seafloor, which rather seems to be more heterogeneously residual, with most of this heterogeneity reliable to previous degrees of melt extraction (see also Warren, 2016).

I might be biased, and I do not want to convince the authors that their model is not correct. But as a pure suggestion I think that the study will represent a more honest and excellent contribution if the message will be reevaluated. For instance, what if the present data are used to re-evaluate the origin of the ophiolite stratigraphy vs present-day MOR stratigraphy? The stratigraphy of most iconic ophiolites (Oman, Cyprus) seems (although we need to drill a 2-km deep hole to find out) different from the present-day MOR. I am leading an IODP effort to drill what will be the deepest hole into the oceanic mantle (IODP Proposal 971Full2), and I can assure that this would be a far-reaching message for the entire Earth Science community. A hypothesis to test to further sustain deep drilling.

Our response: These comments are very helpful for our revision (lines 222-227, 245-249, 259-270, and 318-334). The discussion related to the Vema lithospheric section (Brunelli et al., 2018) in the former version has been revised to focus on the difference between the present-day oceanic mantle (represented by abyssal peridotites) and the ophiolitic mantle (lines 318-334). Please also see our above responses. We are glad and expect to see a new IODP project aiming to drill much deeper mantle in the future, to collect several-km-deep profiles and to test the observations from ophiolites.

Meanwhile, we totally agree that ancient ultra-depleted components exist in the asthenosphere and oceanic lithospheric mantle, mainly based on Nd-Hf-Os isotopic evidence (e.g., Harvey et al., 2006; Liu et al., 2008; Stracke et al., 2011; Rampone & Hofmann, 2012; Sanfilippo et al., 2019). These ancient components commonly have refractory compositions and harzburgitic-dunitic lithologies with much higher solidi than lherzolites, leading to their limited participation in the decompressional melting of asthenosphere under spreading centers. In addition, their size and distribution in the oceanic

mantle are highly debated, as either widespread small components (e.g., Liu et al., 2008; Stracke et al., 2019) or locally distributed large heterogeneities (e.g., O'Reilly et al., 2009). This will result in their uncertainties about their roles and consequences in the asthenosphere upwelling and melt extraction beneath spreading centers. If their sizes are small enough (e.g., sub-specimen scale) and equally distributed in the upwelling asthenosphere, the ancient ultra-depleted components will not affect the first-order upward depletion of oceanic uppermost lithospheric mantle. If their sizes are too large as proposed by O'Reilly et al. (2009) or their distribution is highly heterogeneous, we will not observe the vertical gradual variations of oceanic uppermost mantle, which have been shown by the Kangjinla case (this study) and other well-preserved ophiolites (e.g., Suhr & Robinson, 1994; Batanova & Sobolev, 2000; Godard et al., 2000). Therefore, for clarity, we only consider a peridotitic DMM source to model the first-order compositional variations (**Figures 3-5** in the revision). Our data and modeling results both suggest that the compositional variations of oceanic uppermost (several-km deep) lithospheric mantle are controlled mainly by melt focusing and interaction in the asthenospheric upwelling column beneath spreading centers (**Figures 5 and 6** in the revision), rather than by decompressional melting or source-mantle heterogeneity (see lines 201-316 in the **Discussion**).

(2) Unexpected depletion in pyroxene trace elements during a process of melt-rock reaction at high permeability.

Melts flowing through high permeability channels (dunite or pyroxene-poor harzburgites) represent a mixture of primary melts produced in deeper parts of the melting region (see Liang and Parmentier, 2010; Spiegelman and Kelemen, 2003; Sanfilippo et al., 2017...). Given that the formation of dunites requires disequilibrium, the melt present in the high permeability body would be more enriched in incompatible elements than the pre-existing mantle, residual in origin. Indeed, the occurrence of Cpx having a MORB-type signature is the reason why high permeability channels were considered melt migration pathways (after Kelemen et al., 1995; Dick and Natland, 1996). Hence, if the mineralogical depletion in the NHZ and NDZ was triggered by interaction between mantle peridotites and melt focused in a km-wide region, the pyroxene must gradually equilibrate with the 'channel melt', increasing their incompatible element compositions as result of mixing and/or AFC processes. The gradual depletion in both WR and in incompatible elements of pyroxene in these rocks rather suggest an open system melting process.

Our response: In order to reproduce both the major- and trace-element compositional variations during the dunitization reaction (pyroxene + silica-poor melt → olivine + silica-rich melt; Kelemen et al., 1995, 1997b), we have carried out new MELTS modeling of open-system reactions between two types of mafic melts and the most fertile lherzolite KJL14-05A. At each mantle P-T combination, melts with each parcel of 4 g are added iteratively to the lherzolite KJL14-05A with an initial mass of 100 g. After the melt-solid bulk system attains thermodynamic equilibration, the producing melts will keep in the bulk system if the melt fraction is below 0.005. Otherwise, the melt portion beyond this threshold will be extracted out of the system. The resultant system with ≤ 0.005 melt fraction is then augmented by a new parcel of 4 g melt, and undergoes a new round of melt-rock equilibration and possible melt extraction. This iteration will stop until the total melt addition reaches 200 g, or until the open-system reaction ceases due to energy limits. The major elements are controlled by energy-constrained phase equilibration at different P-T conditions, while the trace elements are determined by partitioning between solid and melt in each modeling step. The details can be seen in the **Methods** (lines 415-442 in the revision). The modeling parameters and results have been shown in **Supplementary Tables 10-12**. In addition, because the clinopyroxene compositions will be strongly affected by sub-solidus re-equilibration with spinel, olivine and especially orthopyroxene, we therefore only compare the newly-analyzed and modeled whole-rock compositions, instead of mineral compositions (**Figure 5**).

In our modeling, we found that the whole-rock compositional variations (including major oxides and fluid-unaffected, intermediately-slightly incompatible trace elements) of the Kangjinla samples can be well reproduced by the melt-peridotite reactions at 1300 °C in the shallow asthenospheric column under a spreading center, but cannot form at lower mantle temperatures (e.g., 1000 °C and 1100 °C in the lithosphere and 1200 °C in the asthenosphere-lithosphere transition zone; **Figure 5**). The reactions at ~1300 °C with higher melt/rock ratios can result in stronger compositional depletion in major oxides and intermediately-slightly incompatible trace elements until finally dunites are formed, producing the gradual rise in whole-rock MgO and FeO_T as well as lower CaO, Ti and Yb, as observed in the Kangjinla peridotites from the SLZ to the NDZ (**Figure 5**). In contrast, the more incompatible trace elements (e.g., LREE) display only mild enrichment because they are buffered by continuous addition of melts in the modeling (**Supplementary Table 12**), consistent with the LREE patterns of the Kangjinla peridotites (**Supplementary Figure 6**). Therefore, the observed and modeled compositional “depletion” is largely shown by major oxides and intermediately-slightly incompatible trace elements, controlled by phase relations and melt-solid equilibrium during the dunitization reactions. The weak enrichments by continuously added melts can be easily seen in more incompatible elements (e.g., LREE; not including fluid-mobile elements), reflecting the strong effects of bulk partition coefficients, melts compositions and melt-rock ratios.

Note that there is literature proposing that the Lubuosa ophiolites formed in a supra-subduction zone environment, including PGE, Re-Os isotopes and Li and O isotopes (Zhou et al.). Some of these authors proposed that a SSZ stage followed a MOR stage, based on the occurrence of stratiform chromitites and boninite-like magmas. Can you exclude that, rather than melting under a MOR, the sequence suffered a multistage history of melt depletion? (see next comment).

Our response: Previous studies have proposed that the Luobusa ophiolite was firstly produced in a MOR setting followed by subduction-zone metasomatism (e.g., Zhou et al., 1996, 2005) or was totally produced in a forearc spreading center during subduction initiation (e.g., Zhang et al., 2019). However, the lack of subduction-zone-index rocks (e.g., boninite) and the widespread occurrence of MORB-like magmatic rocks make a large group of researchers posit that the Luobusa ophiolite was mainly generated in a mid-ocean ridge (e.g., Wu et al., 2014; Yang et al., 2014; Zhang et al., 2020). Even assuming that the Luobusa ophiolitic mantle has been metasomatized by fluids or hydrous melts in subduction zones or later orogenic belts (affecting the fluid-mobile elements and Li isotopes), the amounts of fluids/melts at lithospheric temperatures are not able to change the major lithospheric-mantle composition and framework that were largely produced in high-temperature mantle processes. Moreover, our model (see lines 297-316 and **Figure 6**) can also be applied to subduction-zone-related spreading centers, as long as the tectonic regime includes plate divergence and asthenosphere upwelling.

In the four zones of the Kangjinla ophiolitic mantle, all whole rocks, Cpx and Opx exhibit consistent left-leaning, depleted REE patterns (**Supplementary Figure 6**), as well as “U-shaped” multi-element patterns with strong enrichments in fluid-mobile elements (e.g., Cs, Rb, Ba, Pb, Li; **Supplementary Figure 7**). The fluid-mobile elements and some light REEs (e.g., La) share similar enrichment extents across the four zones, and show positive correlations with LOI contents (**Supplementary Figures 3, 6, 7**), implying overprinting by secondary fluid metasomatism (including serpentinization) during later subduction-zone or orogenic modifications after the formation of the Kangjinla mantle in a spreading center. This metasomatism did not alter the compositions of major oxides (e.g., MgO, FeO_T, SiO₂, Al₂O₃ and CaO) and intermediately-slightly incompatible elements (e.g., Ti, Yb), as well as the lithological variations from the Kangjinla SLZ to the NDZ. We therefore only use these un-affected elements to reveal the high-temperature mantle processes in the revision.

(3) Applicability of the melting model.

The authors use melting models to show that the WR compositions of the mantle section would require melting degrees ranging from 8 to 20% of a DM mantle, considered too wide to be related to a single melting stage. However, this inference is based on a model that uses a single, homogeneous mantle source. Present estimates for the DM are based either on MORB or AP (S&S 2004; W&H, 2005), which are used to invert melting process and infer the composition of a homogeneous depleted peridotite. These estimates can be correctly used to infer the general depletion character of a mantle peridotite, but not necessarily the chemical depletion occurred during the last melting event. Indeed, Nd-Hf-Os isotopes of AP sometimes reveal a long-term, ancient chemical depletion extending by far (for Os and Hf) the DM estimates (see Liu et al, 2006; Harvey et al, 2010; Stracke et al., 2011; Byerly and Lassiter, 2014...). Likewise, MORB retaining depleted isotopic compositions are locally reported (Sanfilippo et al., 2019; 2021; Stracke et al., 2019). Large portions of the depleted mantle can be more depleted than the average DM estimates (see comment 1). Hence, we cannot be sure that the depletion signal derives from the last melting stage.

Our response: We acknowledge that the convective mantle is heterogeneous in both lithology and geochemistry, with a major DMM framework mixed with some enriched and ultra-depleted components. In our modeling, we only consider a simple peridotitic DMM source to reproduce the first-order compositional variations, because (1) the melting of volatile- and/or pyroxenite-rich sources will start at deeper levels and play an important role in the low-melting-degree situations which do not determine the major architecture of oceanic lithospheric mantle, and (2) the ultra-depleted components with limited melting during asthenospheric upwelling have uncertain sizes and distributions in the residual mantle. We do not emphasize that the observed compositional depletion must derive from the last decompressional melting, which has been excluded in the revision. Consideration of the ultra-depleted components in our modeling is therefore beyond the scope of this study. Please also see our above response to the major comment (1) from Reviewer#1.

In Lines 251-252, the authors state that the heterogeneous distribution of old depletion signals cannot explain the gradual depletion signatures of this 2 km-thick peridotite section. However, what if the mantle section was initially stratified (being formed under a MOR) and successively melted in a SSZ setting? Influx melting would explain the decrease in REE coupled with anomalous enrichments in Li, Sr, Pb and FME in the Cpx-Opx, and the apparent buffer in LREE in pyroxenes. In addition, melting a previously depleted mantle by influx melting would also explain the occurrence of boninites and chromitites. Again, I am not forcing the authors to change their model, just take into consideration this possibility. Or, at least, exclude a possible multistage mechanism of formation in a SSZ environment, as suggested by other groups.

Our response: The scenario that the mantle section was initially stratified via our proposed melt-mantle interaction under a spreading center and then fluxed by fluids/hydrous melts in a SSZ setting is highly possible. This is not inconsistent with our model. Please see the above second part of response to the major comment (2) from Reviewer#1. In addition, fluid-influx decompressional melting, similar to the anhydrous melting modeled in this study, still cannot reproduce the compositional gradients observed from ophiolites, because of the parallel lowering of the solidus and no shortening of decompressional distance. The high-Cr# chromitites in ophiolitic mantle can also be formed by the interaction with high-Mg and high-Si melts (similar to boninites) derived from the shallow depleted harzburgitic mantle in any spreading centers with high fluxes of melts (no need for subduction zones; e.g., Gonzalez-Jimenez et al., 2014); there also is no evidence for the existence of boninites or other subduction-zone-related basaltic rocks in the Luobusa ophiolite as well (e.g., Wu et al., 2014). Therefore, the subduction-zone metasomatism (if existed) only had a weak and secondary effect on the Luobusa ophiolitic mantle, and did not modify the major compositions and lithologies of the mantle section produced in an oceanic spreading center.

(4) Model of melt focusing.

To sustain a process of melt focusing the authors use a MELTS algorithm that simulates the assimilation of mantle minerals within a melt with two distinct compositions: silica unsaturated and silica saturated. My first concern is why a melt formed at lower melting degrees (i.e., Grt facies) would be silica undersaturated and a melt formed at higher melting degrees (Grt+Sp1) would approach silica saturation. This has almost no significance in the general model, but should be somehow justified.

Our response: We have deleted these statements, and revised them in lines 271-296.

In the same model they also change the temperature, arguing that the process occurred in the asthenosphere. Why the addition of olivine cannot occur at the TBL, leading to a stratification originally parallel to the seafloor? This is proposed by several workers to explain the occurrence of dunites at the crust mantle boundary. For instance, the Niu 2004 model is based on the addition of Ol at the expenses of Px, in a process occurring within the TBL (1100°C). The same process has been reported in several field studies (Kelemen et al., 1992; Rampone et al., 2008; Piccardo et al., 2007...) which argue that the mineralogical depletion of peridotites can occur at lithospheric levels without the addition of plagioclase, through a process of vertical intergranular migration of basaltic melt. The same has been modelled by Collier and Kelemen, who show that olivine is still the first mineral to form during melt interaction at high temperature (1100°C). In essence, if the rocks formed during a process of large-scale melt focusing, why melt migration might not have occurred at a grain boundary scale converting a Py-rich into a Py-poor peridotite at the TBL, and producing an originally horizontally stratified mantle?

Our response: Yes, the dunitization of peridotites (consumption of pyroxene and addition of olivine) can occur in the thermal boundary layer (i.e., lithosphere), as proposed by Niu (2004) and Collier & Kelemen (2010). These phenomena were also observed in ophiolites (e.g., Kelemen et al., 1995; Piccardo et al., 2007; Rampone et al., 2008), suggesting a process of vertical intergranular migration of silica-undersaturated melts within the lithospheric mantle. However, the pervasive migration of melts along grain boundaries from deep to shallow lithospheric mantle (e.g., Niu, 2004) can produce the deeper mantle with more added olivine (high melt/rock ratios) and the shallower mantle with less olivine (low melt/rock ratios), which is the opposite of the upward depletion observed in this study and other ophiolites. If the melts migrate upwards along hydrofracture-controlled weak zones in the lithospheric mantle (e.g., Nicolas, 1986; Kelemen et al., 1997b), they can only form local dunite dykes/lenses cutting the foliation of wall-rock mantle. This also cannot produce a km-scale lithological zonation as shown in the Kangjinla ophiolitic mantle and other well-preserved ophiolites (e.g., Suhr & Robinson, 1994; Batanova & Sobolev, 2000; Godard et al., 2000).

In addition, from the thermodynamic aspect, the asthenosphere-derived melts that migrate upwards within the lithospheric mantle will lose heat rapidly. This will result in the very rapid and early saturation of olivine in the deeper lithospheric levels (e.g., Collier & Kelemen, 2010), and then the major saturation of pyroxenes and plagioclase in the shallower lithospheric mantle (commonly seen in the impregnated abyssal peridotites and some ophiolitic mantle; e.g., Cannat et al., 1995; Dick & Natland, 1996; Tartarotti et al., 2002; Kelemen et al., 2004; Bodinier & Godard, 2014; Warren, 2016). We thus modeled melt-peridotite reactions at 1000, 1100, 1200 and 1300 °C across the conditions of lithospheric mantle to the adiabatic asthenosphere, and the modeling results show that the compositional depletion and the conversion of pyroxene-rich to pyroxene-poor peridotites can be well reproduced only at 1300 °C (**Figure 5**), an accepted temperature for the shallow asthenosphere (e.g., McKenzie et al., 2005; Ligi et al., 2008; Richards et al., 2018). We therefore proposed a model regarding melt focusing and melt-peridotite interaction in the asthenospheric upwelling column as well as mantle flow beneath a spreading center to explain the compositional stratification of the oceanic uppermost lithospheric mantle (**Figure 6**). This model will provide significant new insights into the

origin of oceanic uppermost mantle and the melt extraction processes under spreading centers, and will attract the interests from a large group of geologists, geochemists, geophysicists and other readers.

Detailed comments:

L40-41: This is not entirely true. Milestone studies on the process of mantle melting, melt migration and melt-rock reaction come from ophiolites: Oman (Kelemen et al. 1990; 1995...) BOI (Surh et al., 2000; 2008), Alpine ophiolites (see Rampone and Sanfilippo, 2021 Elements for a review).

Our response: Revised (see lines 38-44 in the revision).

L42-43: Studies on AP lack a systematic sampling of the oceanic lithosphere, for the obvious reason that the deepest hole into the mantle is 200 m at MAR. But transform faults expose complete lithospheric sections sometimes with long time spans (up to 10 Ma at Vema FZ), and show an heterogeneity comparable, if not higher than reported in this study. Sometimes, the same heterogeneity is revealed in a single dredge. Is the 2-km thick section described in this contribution better representative? (see more comments later).

Our response: We have revised this sentence to make it more reasonable. Please see lines 38-44.

L62: I found this results section a bit too long and detailed. What if you describe the data in a brief manner? I imagine the audience of NatComm would like to see only the most important data. Some of the text can go to the supplements along with the figures.

Our response: Revised to be more readable and concise (lines 136-166 in the revision).

L74-76: the possibility that the section formed after a multistage history of depletion (refs 19, 30, 34) is fundamentally important for deciphering the nature of the section. If you want to export your model to present-day ridges (and I think this is the intention) then you must exclude that a multistage history of melt depletion-melt rock reaction occurred.

Our response: Please see our above responses to the major comments (2) and (3) from Reviewer#1.

L96-97: which means that the top (the base originally) of the CHZ is not exposed in Kangjinla? That's why you deviate your sampling strategy to the west? Is the carbonates exposed in Xiangkashan segment too? Not a big deal, just clarify.

Our response: Clarified (lines 94-102).

L117: they are petrographically and chemically similar to?

Our response: Revised (lines 127-130). Because this part only describes petrography, we did not mention the chemical similarity here.

L118: I know what you mean. But I have been studying dunitites for years.. I am not sure that all readers of NatComm are aware of the dunitization effect. Better to explain later on.

Our response: Revised. The first appearance of dunitization is in line 190 of the revision, followed by subsequent explanation of this term in lines 191-192.

L132-133: Only for WR chemical variations. I agree that serpentinization is isochemical on WR, and most of the variations you see in Fig.2 of supplements are related to olivine addition. However, serpentinization clearly modifies the mineral compositions in terms of trace elements (see high La/Ce ratios).

Our response: We have re-evaluated the serpentinization effect on major and trace elements in both whole rocks and minerals in the revision. Please see lines 136-146 and **Supplementary Figure 3**. We suggest that the fluid-mobile elements and a few LREE (e.g., La) have been strongly modified and

controlled by secondary fluid metasomatism. These elements were therefore not used to reveal the high-temperature mantle processes in the revision.

L135: explain here or before, (addition of olivine at the expenses of pyroxene due to melt migration at high porosity?)

Our response: Revised (lines 189-192 in the revision).

L136: this is absolutely fantastic!! I never saw such a clear trends of gradual chemical depletion...fantastic.

Our response: Yes, it is fantastic. It is why we believe our new observations and novel model deserve publication in Nature Communications.

L145: I like the way you present the data, it is clear and honest. But I do not think this is easy to follow for a non-specialist.

Our response: We have largely shortened the descriptions of data, and avoided the unnecessary details and crude numbers (please see lines 136-166 in the revision).

L146: systematic vertical variations in compatible (xxx) and incompatible elements.. that's it!! Great!!

Our response: Revised.

L147: Yes, but with positive La/Ce fractionation. The same features are seen in AP but likely attributed to serpentinization (see for instance Sani et al., 2020, Lithos). Would you comment on this? Also, I see an interesting difference in REE patterns, the depletion is mainly seen in M-HREE whereas LREE are buffered. Samples from SHZ have the same Ce than the other rocks. If you plot Ce/Nd ratios they decrease upsection. Does it imply some higher extent of interaction with melts? Most AP do not plot along a melting path in L/MREE ratios, which for some authors can be related to some interaction with melts (see Warren, 2016). In my view, this is more a matter of source heterogeneity (see later more comments).

Our response: Yes, we also suggest that the enrichments in La and fluid-mobile elements were probably due to later serpentinization or other fluid metasomatism. The mild enrichments in other LREEs may reflect the buffering effect of continuous addition of melts during our proposed melt-peridotite interaction in the asthenospheric upwelling column. We therefore only focused on major elements and intermediately-slightly incompatible trace elements (e.g., Ti, HREE) here. The discussion of the origin of LREE/MREE ratio variations is beyond the scope of this study, and will be investigated in our future contributions.

L172-173: here it is.. What is the reason? Water, influx melting in a SSZ, melt-rock reaction? We do not really see such enrichments in AP, even if they are much more serpentinized and evidently suffered much higher degrees of alteration.

Our response: Please see our above responses to the major comments (2) and (3) from Reviewer#1.

L182-183: this is somehow unusual to me.. dunitization is generally related to mineral depletion but increase in incompatible elements in pyroxene, which are supplied by the melts. This is the base of the finding of Kelemen et al (1995) who found MORB-type Cpx in dunites.. same in dunites from Lanzo (see Sanfilippo et al., 2017 and ref therein). The gradual depletion in REE in pyroxene can be related to a depleted nature of the migrating melt. Maybe formed in a geodynamic scenario not necessarily MOR-like. This would also explain the high Sr-Pb and FME...

Our response: Please see our above responses to the major comments (2) and (3) from Reviewer#1.

L194: How can you apply the same model of Oman ophiolites in such a dismembered section? Was the spreading phase so close to the obduction? If you have data, please provide reference.

Our response: The common obduction of ophiolites from their precursor oceanic lithosphere occurs via thrusting along the ridge axis, resulting in the emplacement of just-produced juvenile oceanic lithosphere onto the opposite plate. In many ophiolites, this thrusting (obduction) occurs within less than 20 Ma after the production of oceanic lithosphere in a spreading center (e.g., Nicolas, 1989). This mechanism can also be suitable for the Luobusa ophiolite; previous geochronological evidence of a large suite of gabbros and dolerites suggests that the Luobusa ophiolite underwent rapid intra-oceanic emplacement after its formation at ~128-131 Ma in an oceanic spreading center (Zhang et al., 2016).

L204: Delete “the” and add “a mantle”.

Our response: Revised (line 186 in the revision).

L206-208: this concept can be anticipated before... in the data presentation. If you have textures that would help a lot (in the supplements).

Our response: Revised. The concept of dunitization has been firstly shown and explained in lines 189-192. New petrographic images have been added in **Supplementary Figure 1**.

L209: “oceanic” to “ophiolitic”.

Our response: We did not revise it here, because most ophiolites represent relics of oceanic lithosphere. In addition, we have discussed the so-called “dichotomy” between MOR and ophiolitic stratigraphy (see lines 318-334 in the revision).

L213-214: this is my main point.. "in the ophiolitic stratigraphy!" Can you transpose this to present-day MOR? Based on the thousands of samples available, pyroxene-poor harzburgites and dunites are a minority. If AP samples the uppermost oceanic mantle, why they do not recall the stratigraphy seen in the ophiolites? There are several locations where the CMB has been exposed at MORB, and harzburgite is a main lithology.

Our response: Please see our above response to the major comment (1) from Reviewer#1.

L235-236: That's right, but to what extent this depleted mantle peridotite can be used as starting point for calculating melting degrees at the ridge axis? (see later).

Our response: At the partial melting degree of ~1.5% where anhydrous melting of peridotite starts (**Figure 6a**; Ligi et al., 2008), the major framework of residual mantle columns begins to form via decompressional melting and melt-peridotite interaction.

L251-252: then the initial mantle was chemically heterogeneous, and chemical depletion might have been partly inherited from old melting events? What do you mean for heterogeneous distribution of ancient melting residues? What if the mantle section was already differently depleted (under a MOR) and successively melted in a SSZ setting? In-flux melting in a hydrous environment would cause depletions in both in WR and pyroxene composition, better than melt-rock reaction. Hydrous melting would also explain the anomalous enrichments in Sr, Pb and FME (including Li) in the depleted rocks, and the buffer in LREE. I am not arguing that my model is correct, just take into consideration this possibility. Or, at least, exclude a multistage history of depletion, which is an idea published by other groups for the same ophiolites.

Our response: Please see our above responses to the major comments (2) and (3) from Reviewer#1.

L264: Please, check references. The Niu 2004 model does not include PI-impregnation, a process relatively rare at MOR and reported at U-SSR or fracture zones (e.g., Tartarotti et al., 2002).

Our response: Revised (lines 259-261 in the revision).

L278-279: why a melt formed at lower melting degrees (i.e., Grt facies) would be silica unsaturated and a melt formed at higher melting degrees (Grt+Spl) would approach silica saturation??

Our response: We have deleted the previous sentences and revised them in lines 259-270.

L291-292: Why? This is something I do not understand. The Niu 2004 model is based on the addition of Ol at the expenses of Px above the TBL (1100°C). As this process has been reported from several field studies (Kelemen et al., 1990; Rampone et al., 2008; Piccardo et al., 2007...) which clearly show that the mineralogical depletion of several peridotites can occur at TBL. The same has been modelled by Collier and Kelemen, who show that dunite is still a main rock-type to form during melt interaction at TBL. Why your model differs from those reported in literature?

Our response: Please see our above response to the major comment (4) from Reviewer#1.

L304-306: the biggest problem here is why the depleted rocks retain incompatible element depletions in pyroxene. Whenever the melt is focussed, the reacted mantle would suffer the addition of Ol at the expenses of Py, but increasing the incompatible trace elements of the pyroxene. The 'channel melt' being more enriched in incompatible elements than the host rocks (Kelemen et al., 1995; Liang Parmentier, 2010). I suggest to try modelling the REE composition of the Cpx, using AFC equations and the same process as modelled with MELTS.

Our response: Please see our above response to the major comment (2) from Reviewer#1.

L326-329: that's right, but they are related to different melting degrees, and not to different extent of melt-rock reaction. In addition, note that if your model is correct, this would imply that the entire mantle section in Vema (few hundreds of Km in thickness) is formed by highly refractory dunites.

Our response: Thanks for this constructive comment. We have deleted the discussion regarding the Vema mantle section, and largely revised this paragraph to focus on the implications about the difference between present-day MOR mantle (revealed by abyssal peridotites) and fossil oceanic lithospheric mantle (revealed by well-preserved ophiolites). Please see lines 318-334 in the revision.

L329-330: You should refer to the recent literature for Vema, where Brunelli et al. 2018 (Nat Geosc) explain the different melting degrees as consequence of mantle heterogeneity.

Our response: Revised.

L340: Which must have a pyroxene with a composition close to the migrating melt rather than a more residual character.

Our response: Yes, we agree that the compositions of clinopyroxenes (if they exist) in the dunite channels are equilibrated with the passing melts, as shown by Kelemen et al. (1995) and Sanfillippo et al. (2017). These clinopyroxenes crystallized when later MORB melts migrated through the previously-formed dunite channels. However, in the Kangjinla case, depleted anomalies enclosing local dunite lenses in the SLZ and the basal CHZ have almost no clinopyroxene preserved. The rare orthopyroxenes show resorbed shapes indicating strong consumption during interaction with silica-undersaturated melts (see the petrographic descriptions in lines 127-134 in the revision). This is the reason why the Kangjinla samples with stronger melt-peridotite reaction show higher extents of compositional "depletion", supported by our modeled results (**Figure 5**).

Best regards,

Alessio Sanfilippo

Associate Professor in Petrology,
Department of Earth and Environmental Science,
University of Pavia,
Via Ferrata, 1, 27100 Pavia, Italy
Tel: +39 0382985789
email: alessio.sanfilippo@unipv.it

Responses to comments from Reviewer #2 (Anonymous)

Reviewer #2 (Remarks to the Author):

Review of Vertical depletion of ophiolitic mantle decodes melt focusing and interaction in the asthenospheric column under oceanic spreading centers

By Qing Xiong, Hong-Kun Dai, Jian-Ping Zheng, William L. Griffin, Hong-Da Zheng, Li Wang, and Suzanne Y. O'Reilly

The authors present new major and trace element results for peridotites from the Kangjinla massif (Luobusa ophiolite), with the specificity of having applied a cross-section sampling strategy from the deepest to shallowest levels of the mantle section (about 2km vertically). A vertical petrological and geochemical evolution is described, with more refractory characters upward, defined by the decrease in the clinopyroxene content and more and more depleted chemical compositions. Based on this observation, the authors performed thermodynamical models to show that this compositional gradient cannot directly result from decompressional melting, but requires melt/rock interaction in the asthenosphere prior to mantle flowing and lithospherization. The data are of good quality and the methods are well detailed. The manuscript and figures are very clear and easily understandable. The proposed model is interesting and elegant to account for the observed spatial chemical variation. However, a few points need to be addressed, discarded or discussed, before making this manuscript suitable for publication in Nature Communications. I detail here below these few points as questions or comments.

(1) Origin of clinopyroxene: residual origin or refertilization?

Clinopyroxene-bearing harzburgites or lherzolites have also been observed in other ophiolites in the deepest levels of mantle sections, close to the thrust contact, such as in the Oman (Godard et al., 2000; Takazawa et al., 2003; Prigent et al., 2018) or Bay of Island (Girardeau and Nicolas, 1981) ophiolites. In some cases clinopyroxene (cpx) is better explained as a result of melt-peridotite reaction, possibly 'at near-solidus conditions along the lithosphere-asthenosphere boundary' (in the abstract of Godard et al., 2000), than as a residual mineral phase.

In your study, you likely observe cpx-bearing peridotites in the deepest SHZ unit. You consider this cpx as a residual phase, making the host rocks your more 'fertile' samples, but you do not really discuss the possibility of a reactive origin. It should be done, at least in one paragraph in the discussion, as it is a critical point to support your following interpretation - 'the most fertile harzburgite (KJL14-05A)' (line 273) is used for modelling. In the same way, please better describe the texture of cpx in the results section as it is now not detailed excepting 'sometimes occurring as porphyroblasts' (Lines 109-110) (associated with Opx?). A good argument supporting the residual origin, and you mention it several times, is that the cpx disappears upward gradually, i.e. it is not restricted to the vicinity of the few hundred meters at the base of SHZ close to Langjiexue Group (Fig 1), but it should be stated if it is your main argument to favour the residual vs. refertilization origin.

This is the main criticism I have as if the residual origin of cpx is not clearly proven, and finally is a melt-rock reaction/refertilization product, all the following discussion-interpretation fall.

Our response: Thanks for this constructive comment. We have added more petrographic descriptions (lines 119-124) and discussion (lines 272-278) of lherzolites in the SLZ (especially for the sample KJL14-05A) in the revision. All the lines of evidence support that the lherzolite KJL14-05A represents the most fertile end-member with a residual origin, which has only been weakly overprinted by cryptic metasomatic enrichments in some highly incompatible elements (not affect its major primary compositions). It is also petrographically and geochemically similar to the Zedang lherzolites (Xiong

et al., 2017) and the Oman Type-I lherzolites (Takazawa et al., 2003), both consistent with a residual origin in decompressionally-melted asthenosphere. We therefore interpret that the lherzolite KJL14-05A and its porphyroblastic clinopyroxenes have a residual origin, making our modeling and discussion reasonable.

(2) Differences between your cross-sections?

In Figure 1 we can see that you sampled two (sub-)parallel cross-sections across the NHZ and CHZ units. However, there is no distinction in the chemical plots. A slight weakness of your work is that it is based on a solely synthetic cross-section, so we cannot really envision if the global depletion upsection you describe applies to the whole ophiolite or not, while several cross-sections could have shown that. A way to partially overcome this could be to make the distinction between the different cross-sections in your plots, to show if the data from respective cross-section complement each other to form a common trend, or if it is more variable/complicated at small-scale.

Our response: This is a very helpful suggestion. Samples from the nearly parallel cross-sections in the Kangjinla NHZ and CHZ have been collected, aiming to fully cover the lithological gradation from the top of the NHZ to the bottom of the CHZ (**Figure 1c**). We have plotted the compositional variations as separate cross-sections in the NHZ and CHZ, according to this suggestion (see the following **Figure R1**). They still show the same first-order variation trends as those in the reconstructed single profile (**Figure R1 and Figure 2**). We therefore keep the treatment to use all the Kangjinla samples in a single reconstructed mantle profile in the revision.

Figure R1. North-south variations of partial-melting degree F (a), whole-rock MgO (b), orthopyroxene Cr# (c), clinopyroxene Mg# (d), whole-rock Ti_N (normalized to PM; e) and clinopyroxene Yb_N (PM; f) of peridotites from the four zones of the Kangjinla ophiolitic mantle. The different cross-sections (#1 and #2 in the NHZ, and #3 and #4 in the CHZ) are shown, and display the same trends as those of the reconstructed single profile in the revision. The abbreviations see those of **Figure 2** in the revision.

(3) High-density sampling, and continuity between units.

Another point is that you did not sampled the base of the CHZ just above the SHZ - there is no direct continuity in the sampling from the cpx-rich SHZ to the cpx-poor or cpx-free CHZ while there is a clear, distinct chemical shift between these two units (Fig. 2). Accordingly, there is a gap of a few hundred meters between the top of SHZ near 1750 m and the bottom of CHZ near 1350 m (Fig. 2 and

Suppl Table 1). Are there any chemical data from literature that could fill this gap and expand/confirm the general trend you describe? Surprisingly you compare your data with compositions of abyssal peridotites, ok with that, but not with compositions of previous works on the same Luobusa/ Kangjinla ophiolite.

Our response: It is a bit regrettable that we did not have the sample coverage from the bottom of the CHZ to the top of SLZ in this study, with a roughly 400-m distance either totally covered by Triassic black slates or strongly altered by carbonation and serpentinization. There are no published relevant data yet in our un-sampled section, and we are the first to carry out such a high-resolution and systematic work on the whole Kangjinla ophiolitic mantle section. Despite all this, we collected all published reliable whole-rock data of peridotites from different localities in the Luobusa ophiolite (without spatial constraints as our sampling strategy) in the revision, and compared them to our new data which have good spatial constraints (**Figures 3 and 5** in the revision). Our data share the same compositional range of the literature data, further testing our observations.

An additional observation is that you report dunites at the base of the CHZ (Suppl Fig. 1). It is also commonly observed, together with the presence of cpx-bearing harzburgites, just above the basal thrust contact of ophiolites (e.g. in Oman, Klaessens et al., 2021). These basal dunites are usually interpreted as formed during the intra-oceanic thrusting leading to the ophiolite's obduction. If so, how to be sure about the continuity from the SHZ and CHZ, especially regarding the lack of samples and data on 400 m in thickness at the base of CHZ? Could the thrust fault have developed just below the CHZ in the eastern part of the massif? (i.e. just to the south of your blue samples in Fig. 1c).

Our response: The Triassic black slates are indeed thrust above both the SLZ and the CHZ in different segments of the Luobusa ophiolite (**Figure 1c**; Liang et al., 2011), suggesting that the ophiolite and the sediments were juxtaposed during later orogenic assembly. Our and previous studies of the Luobusa ophiolite have never observed the basal mantle layer with foliated and even mylonitic structures and the existence of metamorphic soles related to inter-plate thrusting during ophiolite obduction, as observed in the Oman ophiolite (e.g., Prigent et al., 2018). The basal SLZ and the enclosed dunite lenses in this study show clearly porphyroblastic and ductile microstructures, a typical lithospheric mantle feature produced by asthenosphere upwelling and high-temperature mantle deformation. The observed gradual petrographic and compositional variations of the Kangjinla mantle profile also exclude that the thrust fault played a significant role in the production of the SLZ and its local dunite lenses.

(4) Tilt of the Kangjinla ophiolitic units?

Lines 83-84: 'This zonation reveals that the mantle section was overturned during its emplacement 19,26'. It seems there is a petrological logic in the successive units you describe. However, a critical point not addressed in your manuscript is if the cross-section is really vertical, perpendicular to the paleo-Moho/paleo contacts, or not. What I mean is that if there is an angle between your sampling line and the paleo contacts, it could allow to under/over-estimate the observed 'vertical' depletion (12% as indicated in the abstract Lines 22-23), and accordingly bias the melting pressure range (5 kbars) and thus the depth/thickness (15 km) on which partial melting potentially occurred (Line 239) and estimated from your thermodynamical modelling. You finally conclude that it is unlikely that the data reflect partial melting event as you favour the melt/peridotite processes, but such estimation in the depletion % and/or pressure could be biased. How to be sure that your sampling overcome or took into account a possible structural tilt of the units?, and thus how to be sure your sampling represent 'a high-resolution view of the lithological and compositional variations in a section of the oceanic lithospheric mantle' (Lines 58-59)?

Our response: Thanks for this constructive comment. We have considered the tilt effect on the mantle section thickness in the revision (lines 97-101, 231-235). In fact, the direction of our sampling lines is

generally N-S, perpendicular to the suggested ~E-W strike of the lithospheric stratigraphy (**Figure 1c**; e.g., Zhou et al., 2005). The field geology and geophysical observations also suggest that the whole Luobusa ophiolite section tilts at high angles ($> \sim 45^\circ$) relative to the exposed surface, sometimes with almost perpendicular exposure of the mantle stratigraphy (e.g., Zhou et al., 2005; Jiang et al., 2015). In theory, the exposed distance of the N-S mantle profile represents the maximum estimate of the real mantle section thickness (i.e., the maximum depth of the reconstructed lithospheric mantle). It means that the “melt-depletion” gradient ($\sim 6\%/km$) of the Kangjinla mantle profile calculated using the exposed N-S distance as the mantle depth represents the minimum estimate, which is still significantly higher than those of modeled residual mantle columns (0.44-0.22 $\%/km$) at T_p of 1300-1450 $^\circ C$ (**Supplementary Table 11**). Therefore, whether to calibrate the structural tilt or not does not affect the main inference in this study that decompressional melting cannot produce the vertical depletion features of oceanic uppermost mantle.

(5) Consistency between the model and geological observations?

Lines 314-310: ‘When this column rises it splits into two symmetrical parts, which will dynamically rotate $\sim 90^\circ$ in the mantle-flow regime to become the juvenile uppermost lithospheric mantle^{3,7}, with the primary middle part of the column at the top and the distal part to the bottom (Fig. 6c). This perpendicular rotation of the split asthenospheric column can thus explain the vertical gradient in depletion and the local depleted anomalies in oceanic uppermost lithospheric mantle and ophiolitic mantle.’

This is an elegant model. At the same time, it has been observed both in present-day oceans along oceanic spreading centres (e.g. Dick and Natland, 1996; Godard et al.; 2008) and in un-transposed ophiolitic massifs (e.g. Maqсад area in Oman, Rabinowicz et al., 1987; Abily and Ceuleneer, 2013; Rospabé et al., 2017) that the transition between the mantle section and the base of the crust is extensively made of dunites. This is what you describe for the NHZ, so it’s tempting to interpret this unit as the original (paleo-)Moho. How to account this observation with the idea that the deepest mantle section has been rotated by 90° ? A decoupling between the NHZ and other units, while the gradational chemical evolution over all units is observed?

Our response: We speculate that what you mean is the NDZ (not the NHZ) decoupled with other units, if it can represent the dunite transition zone as a paleo-Moho (e.g., Rabinowicz et al., 1987; Dick & Natland, 1996; Godard et al., 2008; Abily & Ceuleneer, 2013; Rospabé et al., 2017). The NDZ in this study is actually not equal to the whole dunite transition zone described in the Oman case, but is just similar to the lower part of the Oman dunite transition zone, made up of mainly reactive dunites and relict harzburgites (Abily & Ceuleneer, 2013; Rospabé et al., 2017). The upper part of the dunite transition zone has been interpreted to have a cumulative origin (e.g., Abily & Ceuleneer, 2013) and to be produced at lithospheric Moho depths (on top of the melt-focusing center of the asthenospheric upwelling column). Therefore, there is no dynamic decoupling among the four zones in the Kangjinla mantle section. We proposed a model that the compositional variations formed in the giant asthenospheric upwelling column which then split and rotated $\sim 90^\circ$ to form the horizontal uppermost lithospheric mantle. The generation of the cumulative lowermost crust and the following crustal magmatic evolution occurred in the shallow levels and not in the mantle flowing zone under spreading centers.

Additional comments:

Line 26: ‘melt/rock variations’. “melt/rock ratio variations” would be clearer.

Our response: Revised (see line 25 in the revision).

Lines 49-52: 'For example, the fractal dunite melt-channel system in the upwelling residual mantle has been revealed by studies of ophiolites (mainly the Oman example) to illustrate the melt-extraction processes and mantle dynamics under oceanic spreading centers^{6,24}'. Has been 'revealed' is not exactly the proper term. The fractal channel has not been directly observed in ophiolites, especially Oman, but was conceptualised from the observation of horizontal channels possibly transposed off-axis, which are much more common in ophiolitic sequences than vertical channels - see the Figure 16 caption in Braun and Kelemen 2002 (the black box in the top right corner that locates what was really observed on the field). The fractal concept has actually been more supported by numerical modelling (theory) than by strong field observations (practice). It is not a big point here, but maybe simply replace 'revealed' by 'conceptualised', 'theorised', 'proposed', or another word.

Our response: Revised as "proposed" as suggested (see line 50 in the revision).

Lines 105-106: 'porphyroblastic textures and plastic deformation'. This short description supports mantle flowing related to off-axis transposition of asthenospheric mantle described in your model. However, we cannot really check that on the four pictures in Suppl Fig 1 as we do not see mineral preferential orientations or elongated minerals. Could it be possible to add additional pictures? Are there different deformation features between samples from NHZ on one hand, and other samples on the other hand? (see my comment about NHZ above).

Our response: We have revised and added more descriptions of the sample petrology (lines 104-134 in the revision) and new petrographic photos in **Supplementary Figure 1**. We wish that these modifications can provide solid evidence to support our model.

Lines 326-332: 'However, within the same sampling site on a spreading center, such as the narrow ~22-24 Ma sector in the Vema Fracture Zone of the Mid-Atlantic Ridge, spinel Cr# values of the abyssal peridotites also show a large range, indicating major compositional variations⁵⁹. The large variations are difficult to explain by the almost consistent physiochemical conditions and sources within this small domain, but may be easily resolved by the sampling of a short profile of the vertical uppermost lithospheric mantle produced by the processes proposed in this study (Fig. 6).' El Dien et al. (2019) recently shown that the chemistry of Cr-spinel could possibly reflect metasomatism only, and not to be a mantle melting indicator.

Our response: Thanks for this comment. This part has been deleted and revised to focus on the implications about the difference between present-day MOR mantle (revealed by abyssal peridotites) and fossil oceanic lithospheric mantle (revealed by well-preserved ophiolites). Please see lines 318-334 in the revision. We therefore do not need to discuss the Cr# of spinel reflecting possible metasomatism proposed by El Dien et al. (2019).

The methods are well presented and detailed. Values for a few standards are given in Suppl Table 2 following the whole rock major element compositions. You also mention in the text having analysed multiple standards during laser ablation analyses (Lines 385-386). Mean values of repeated analyses of these standards could be given in Suppl Table 7 or 8 after clinopyroxene or orthopyroxene trace element data.

Our response: Standard results have been added in **Supplementary Tables 2, 3, 8 and 9**.

I hope my comments/questions will help you to clarify some points of your manuscript.

Best regards

Responses to comments from Reviewer #3 (Dr. Dongyang Lian)

Reviewer #3 (Remarks to the Author):

Dear Editors and Authors:

Thanks for giving me the chance to review the manuscript NCOMMS-22-07539 “Vertical depletion of ophiolitic mantle decodes melt focusing and interaction in the asthenospheric column under oceanic spreading centers”.

I have read this manuscript with great interest. The authors have presented us a whole picture on the vertical compositional variations of oceanic mantle in a spreading center, showing upward depletion and local depleted anomalies. In general, this manuscript is well written, I would suggest acceptance with a minor revision for this manuscript. I only have some minor comments as listed below:

(1) Line 39-42: The authors suggest that oceanic crust can be studied through research on fossil ophiolites, but the lithospheric mantle is mainly studied from abyssal peridotites, which is contradictory. There are plenty of studies on oceanic mantle from global ophiolitic peridotites.

Our response: Thanks for this comment. We have revised this paragraph (lines 38-44 in the revision).

(2) Line 63: Normally this section should go immediately after the "introduction", rather than as a part of the "result" section.

Our response: We keep this part in the **Results**, because the descriptions of geology and petrology are important components of the results in our study.

(3) Line 66-68: I would suggest the authors to put some examples of peridotites from the northern sub-belt of YZSZ to give the readers a whole picture on the YZSZ ophiolites, and put the related references here.

Our response: We have added relevant references (line 66 in the revision).

(4) Line 131-133: It seems unreasonable that serpentinization will not affect the composition of peridotites.

Our response: We have revised this paragraph to show the effects of serpentinization on the compositions of the Kangjinla peridotites (lines 136-146 in the revision).

(5) Line 154, 158: Please verify the number of significant digits here.

Our response: Revised (lines 147-166 in the revision).

(6) Line 292-295: It is confusing for me here. How can silica saturated melts reacting with mantle peridotites to produce the more depleted endmembers. It is not clear here what is the different roles played by these different types of magma in the generation of the chemical trend of the Kanjingla peridotites.

Our response: We have revised this confusing part (lines 259-270 in the revision), and reorganized the descriptions of the thermodynamic modeling in the **Methods**. It is actually the silica-undersaturated (not silica-saturated) melts reacting with peridotitic mantle that can produce the depleted compositions (**Figure 5** in the revision).

References in the Revision Notes:

- Abily, B. & Ceuleneer, G. The dunitic mantle-crust transition zone in the Oman ophiolite: Residue of melt-rock interaction, cumulates from high-MgO melts, or both? *Geology* **41**, 67-70 (2013).
- Batanova, V. G. & Sobolev, A. V. Compositional heterogeneity in subduction-related mantle peridotites, Troodos massif, Cyprus. *Geology* **28**, 55-58 (2000).
- Bodinier, J. L. & Godard, M. Orogenic, ophiolitic, and abyssal peridotites. in *Treatise on Geochemistry 2nd Edition* (eds Holland, H. D. & Turekian, K. K.) **3**, 103-167 (Elsevier, 2014).
- Braun, M. G. & Kelemen, P. B. Dunite distribution in the Oman ophiolite: implications for melt flux through porous dunite conduits. *Geochem. Geophys. Geosyst.* **3**, 1-21 (2002).
- Brunelli, D., Cipriani, A. & Bonatti, E. Thermal effects of pyroxenites on mantle melting below mid-ocean ridges. *Nature Geosci.* **11**, 520-525 (2018).
- Byerly, B. L. & Lassiter, J. C. Isotopically ultradepleted domains in the convecting upper mantle: Implications for MORB petrogenesis. *Geology* **42**, 203-206 (2014).
- Cannat, M., Karson, J. A., Miller, D. L. & other Shipboard Scientific Party. Site 920. *Proc. ODP. Init. Rep.* **153**, 45-119 (1995).
- Collier, M. L. & Kelemen, P. B. The case for reactive crystallization at mid-ocean ridges. *J. Petrol.* **51**, 1913-1940 (2010).
- Dick, H. J. B. & Natland, J. H. Late-stage melt evolution and transport in the shallow mantle beneath the East Pacific Rise. *Proc. ODP. Sci. Rep.* **147**, 103-134 (1996).
- Dick, H. J. B., Lin, J. & Schouten, H. An ultraslow-spreading class of ocean ridge. *Nature* **426**, 405-412 (2003).
- Dick, H. J. B. & Zhou, H. Y. Ocean rises are products of variable mantle composition, temperature and focused melting. *Nature Geosci.* **8**, 68-74 (2015).
- El Dien, H. G. et al. Cr-spinel records metasomatism not petrogenesis of mantle rocks. *Nat. Comm.* **10**, 5103, doi: 10.1038/s41467-019-13117-1 (2019).
- Girardeau, J. & Nicolas, A. The structures of two ophiolite massifs, Bay-Of-Islands, Newfoundland: A model for the oceanic crust and upper mantle. *Tectonophysics* **77**, 1-34 (1981).
- Godard, M., Jousset, D. & Bodinier, J. L. Relationships between geochemistry and structure beneath a palaeo-spreading centre: a study of the mantle section in the Oman ophiolite. *Earth Planet. Sci. Lett.* **180**, 133-148 (2000).
- Godard, M., Lagabrielle, Y., Alard, O. & Harvey, J. Geochemistry of the highly depleted peridotites drilled at ODP Sites 1272 and 1274 (Fifteen-Twenty Fracture Zone, Mid-Atlantic Ridge): Implications for mantle dynamics beneath a slow spreading ridge. *Earth Planet. Sci. Lett.* **267**, 410-425 (2008).
- González-Jiménez, J. M. et al. Chromitites in ophiolites: How, where, when, why? Part II. The crystallization of chromitites. *Lithos* **189**, 140-158 (2014).
- Harvey, J. et al. Ancient melt extraction from the oceanic upper mantle revealed by Re-Os isotopes in abyssal peridotites from the Mid-Atlantic ridge. *Earth Planet. Sci. Lett.* **244**, 606-621 (2006).
- Harvey, J. et al. Unravelling the effects of melt depletion and secondary infiltration on mantle Re-Os isotopes beneath the French Massif Central. *Geochim. Cosmochim. Acta* **74**, 293-320 (2010).
- Jiang, M. et al. Seismic reflection and magnetotelluric profiles across the Luobusa ophiolite: Evidence for the deep structure of the Yarlung Zangbo suture zone, southern Tibet. *J. Asian Earth Sci.* **110**, 4-9 (2015).
- Kelemen, P. B. Reaction between ultramafic rock and fractionating basaltic magma I. Phase relations, the origin of calc-alkaline magma series, and the formation of discordant dunite. *J. Petrol.* **31**, 51-98 (1990).
- Kelemen, P. B., Dick, H. J. B. & Quick, J. E. Formation of harzburgite by pervasive melt/rock reaction in the upper mantle. *Nature* **358**, 635-641 (1992).
- Kelemen, P. B., Shimizu, N. & Salters, V. J. M. Extraction of mid-ocean-ridge basalts from the upwelling mantle by focused flow of melt in dunite channels. *Nature* **375**, 747-753 (1995).
- Kelemen, P. B., Koga, K. & Shimizu, N. Geochemistry of gabbro sills in the crust-mantle transition zone of the Oman ophiolite: implications for the origin of the oceanic lower crust. *Earth Planet. Sci. Lett.* **146**, 475-488 (1997a).
- Kelemen, P. B., Hirth, G., Shimizu, N., Spiegelman, M. & Dick, H. J. A review of melt migration processes in the adiabatically upwelling mantle beneath oceanic spreading ridges. *Phil. Trans. R. Soc. Lond. A* **355**, 283-318 (1997b).
- Kelemen, P. B., Kikawa, E., Miller, D. J. & other Shipboard Scientific Party. Leg 209 summary. *Proc. ODP. Init. Rep.* **209**, 1-139 (2004).
- Klaessens, D., Reisberg, L., Jousset, D., Godard, M. & Aupart, C. Osmium isotope evidence for rapid melt migration towards the Moho in the Oman ophiolite. *Earth Planet. Sci. Lett.* **572**, 117111 (2021).

- Liang, Y. & Parmentier, E. M. A two-porosity double lithology model for partial melting, melt transport and melt-rock reaction in the mantle: Mass conservation equations and trace element transport. *J. Petrol.* **51**, 125-152 (2010).
- Liang, F. H. et al. Tectonic occurrence and emplacement mechanism of ophiolites from Luobusa-Zedang. *Acta Petrol. Sin.* **27**, 3255-3268 (2011, in Chinese with English abstract).
- Ligi, M., Cuffaro, M., Chierici, F. & Calafato, A. Three-dimensional passive mantle flow beneath mid-ocean ridges: an analytical approach. *Geophys. J. Int.* **175**, 783-805 (2008).
- Liu, C. Z. et al. Ancient, highly heterogeneous mantle beneath Gakkel ridge, Arctic Ocean. *Nature* **452**, 311-316 (2008).
- McKenzie, D., Jackson, J. & Priestley, K. Thermal structure of oceanic and continental lithosphere. *Earth Planet. Sci. Lett.* **233**, 337-349 (2005).
- Nicolas, A. A melt extraction model based on structural studies in mantle peridotites. *J. Petrol.* **27**, 999-1022 (1986).
- Nicolas, A. Structures of ophiolites and dynamics of oceanic lithosphere. in *Petrology and structural geology* (ed. Nicolas, A.), **4**, 1-367 (Kluwer Academic Publishers, 1989).
- Niu, Y. L. Bulk-rock major and trace element compositions of abyssal peridotites: Implications for mantle melting, melt extraction and post-melting processes beneath mid-ocean ridges. *J. Petrol.* **45**, 2423-2458 (2004).
- O'Reilly, S. Y., Zhang, M., Griffin, W. L., Begg, G. & Hronsky, J. Ultradeep continental roots and their oceanic remnants: A solution to the geochemical "mantle reservoir" problem? *Lithos* **211S**, 1043-1054 (2009).
- Piccardo, G. B., Zanetti, A. & Muntener, O. Melt/peridotite interaction in the Southern Lanzo peridotite: Field, textural and geochemical evidence. *Lithos* **94**, 181-209 (2007).
- Prigent, C., Agard, P., Guillot, S., Godard, M. & Dubacq, B. Mantle wedge (de)formation during subduction infancy: Evidence from the base of the Semail ophiolitic mantle. *J. Petrol.* **59**, 2061-2092 (2018).
- Rabinowicz, M., Ceuleneer, G. & Nicolas, A. Melt segregation and flow in mantle diapirs below spreading centers: evidence from the Oman ophiolite. *J. Geophys. Res.: Solid Earth* **92**, 3475-3486 (1987).
- Rampone, E., Piccardo, G. B. & Hofmann, A. W. Multi-stage melt-rock interaction in the Mt. Maggiore (Corsica, France) ophiolitic peridotites: microstructural and geochemical evidence. *Contrib. Mineral. Petrol.* **156**, 453-475 (2008).
- Rampone, E. & Hofmann, A. W. A global overview of isotopic heterogeneities in the oceanic mantle. *Lithos* **148**, 247-261 (2012).
- Rampone, E. & Sanfilippo, A. The heterogeneous Tethyan oceanic lithosphere of the Alpine ophiolites. *Elements* **17**, 23-28 (2021).
- Richards, F. D., Hoggard, M. J., Cowton, L. R. & White, N. J. Reassessing the thermal structure of oceanic lithosphere with revised global inventories of basement depths and heat flow measurements. *J. Geophys. Res.: Solid Earth* **123**, 9136-9161 (2018).
- Rospabé, M., Ceuleneer, G., Benoit, M., Abily, B. & Pinet, P. Origin of the dunitic mantle-crust transition zone in the Oman ophiolite: The interplay between percolating magmas and high-temperature hydrous fluids. *Geology* **45**, 471-474 (2017).
- Salters, V. J. M. & Stracke, A. Composition of the depleted mantle. *Geochem. Geophys. Geosyst.* **5**, Q05004, doi:10.1029/2003GC000597 (2004).
- Sanfilippo, A., Tribuzio, R., Ottolini, L. & Hamada, M. Water, lithium and trace element compositions of olivine from Lanzo South replacive mantle dunites (Western Alps): New constraints into melt migration processes at cold thermal regimes. *Geochim. Cosmochim. Acta* **214**, 51-72 (2017).
- Sanfilippo, A., Salters, V., Tribuzio, R. & Zanetti, A. Role of ancient, ultra-depleted mantle in Mid-Ocean-Ridge magmatism. *Earth Planet. Sci. Lett.* **511**, 89-98 (2019).
- Sanfilippo, A., Salters, V. J. M., Sokolov, S. Y., Peyve, A. A. & Stracke, A. Ancient refractory asthenosphere revealed by mantle re-melting at the Arctic Mid Atlantic Ridge. *Earth Planet. Sci. Lett.* **566**, 116981 (2021).
- Sani, C. et al. Ultra-depleted melt refertilization of mantle peridotites in a large intra-transform domain (Doldrums Fracture Zone; 7-8°N, Mid Atlantic Ridge). *Lithos* **374-375**, 105698 (2020).
- Spiegelman, M. & Kelemen, P. B. Extreme chemical variability as a consequence of channelized melt transport. *Geochem. Geophys. Geosyst.* **4**, 1055, doi:10.1029/2002GC000336 (2003).
- Stracke, A. et al. Abyssal peridotite Hf isotopes identify extreme mantle depletion. *Earth Planet. Sci. Lett.* **308**, 359-368 (2011).
- Stracke, A., Genske, F., Berndt, J., Koornneef, J. M. Ubiquitous ultra-depleted domains in Earth's mantle. *Nature Geosci.* **12**, 851-855 (2019).
- Suhr, G. & Robinson, P.T. Origin of mineral chemical stratification in the mantle section of the Table Mountain massif (Bay of Islands Ophiolite, Newfoundland, Canada). *Lithos* **31**, 81-102 (1994).

- Suhr, G., Seck, H. A., Shimizu, N., Gunther, D. & Jenner G. Infiltration of refractory melts into the lowermost oceanic crust: evidence from dunite- and gabbro-hosted clinopyroxenes in the Bay of Islands Ophiolite. *Contrib. Mineral. Petrol.* **131**, 136-154 (1998).
- Suhr, G. & Edwards, S. J. Contrasting mantle sequences exposed in the Lewis Hills massif: Evidence for the early, arc-related history of the Bay of Islands ophiolite. in *Ophiolites and Oceanic Crust: New Insights from Field Studies and the Ocean Drilling Program: Boulder, Colorado* (eds Dilek, Y., Moores, E. M., Elthon, D. & Nicolas, A.). *Geol. Soc. Amer. Spec. Paper* **349**, 433-442 (2000).
- Suhr, G., Hellebrand, E., Snow, J. E., Seck, H. A. & Hofmann, A. W. Significance of large, refractory dunite bodies in the upper mantle of the Bay of Islands Ophiolite. *Geochem. Geophys. Geosyst.* **4**, 8605, doi:10.1029/2001GC000277 (2003).
- Takazawa, E., Okayasu, T. & Satoh, K. Geochemistry and origin of the basal lherzolites from the northern Oman ophiolite (northern Fizh block). *Geochem. Geophys. Geosyst.* **4**, 1021, doi:10.1029/2001GC000232 (2003).
- Tartarotti, P., Susini, S., Nimis, P. & Ottolini, L. Melt migration in the upper mantle along the Romanche Fracture Zone (Equatorial Atlantic). *Lithos* **63**, 125-149 (2002).
- Urann, B. M., Dick, H. J. B., Parnell-Turner, R. & Casey, J. F. Recycled arc mantle recovered from the Mid-Atlantic Ridge. *Nat. Comm.* **11**, 3887, doi:10.1038/s41467-020-17604-8 (2020).
- Warren, J. M. Global variations in abyssal peridotite compositions. *Lithos* **248-251**, 193-219 (2016).
- Workman, R. K. & Hart, S. R. Major and trace element composition of the depleted MORB mantle (DMM). *Earth Planet. Sci. Lett.* **231**, 53-72 (2005).
- Wu, F. Y. et al. Yarlung Zangbo ophiolite: A critical updated view. *Acta Petrol. Sin.* **30**, 293-325 (2014, in Chinese with English abstract).
- Yang, J. S., Robinson, P. T. & Dilek, Y. Diamonds in ophiolites: a little-known diamond occurrence. *Elements* **10**, 123-126 (2014).
- Zhang, C., Liu, C. Z., Wu, F. Y., Zhang, L. L. & Ji, W. Q. Geochemistry and geochronology of mafic rocks from the Luobusa ophiolite, South Tibet. *Lithos* **245**, 93-108 (2016).
- Zhang, P. F. et al. Evolution of nascent mantle wedges during subduction initiation: Li-O isotopic evidence for the Luobusa ophiolite, Tibet. *Geochim. Cosmochim. Acta* **245**, 35-58 (2019).
- Zhang, C., Liu, C. Z., Liu, T. & Wu, F. Y. Evolution of mantle peridotites from the Luobusa ophiolite in the Tibetan Plateau: Sr-Nd-Hf-Os isotope constraints. *Lithos* **362-363**, 105477 (2020).
- Zhou, M. F., Robinson, P. T., Malpas, J. & Li, Z. J. Podiform chromitites in the Luobusa ophiolite (Southern Tibet): Implications for melt-rock interaction and chromite segregation in the upper mantle. *J. Petrol.* **37**, 3-21 (1996).
- Zhou, M. F., Robinson, P. T., Malpas, J., Edwards, S. J. & Qi, L. REE and PGE geochemical constraints on the formation of dunites in the Luobusa ophiolite, southern Tibet. *J. Petrol.* **46**, 615-639 (2005).

Reviewer #1 (Remarks to the Author):

Dear Editor. I have read the new version of the manuscript with interest, following the response to my review as provided by the authors. I admit that they considered my concerns and modified some parts of the manuscript accordingly. In the response letter, they convincingly reply to the main arguments and provide a new revised version of the chemical model to account for the trace element composition of the rocks. On this basis, they state that the paper is now suitable for publication in your journal.

The main reason for my rejection in the first version is that, in my view, this paper does not provide a model applicable to the oceanic lithosphere, because the stratification that they see in the Lubosa ophiolite is not what we expect to be seen in the abyssal mantle. This argument is still there. They argue that abyssal peridotites represent a pelicular portion of the mantle, whereas km-thick oceanic sequences can only be studied at ophiolites. Still, the variability seen at MOR is not comparable to what provided here. The first 200 m of mantle of Lubosa ophiolites are mostly dunites, with some rare harzburgites or troctolites veins (although no analyses are provided). But dunites are rare at MOR, and if you plot the compositional variability of the dredged and drilled samples at present day ocean, it would cover entirely that of this 2 km-thick section. Even if "pelicular", the mantle manifests himself at MOR as an extremely variability assemblage of residual lithologies, variably interacted with melts at high and low T, at high at low P, locally refertilized, impregnated and intruded by gabbroic materials and veins with different composition. Such a variability is a first order process and indicates that refertilization, rather than extreme depletion, is a common characteristic of AP. And such a huge amount of refertilized lithologies are not seen in the Lubosa ophiolites and cannot survive in a huge vertical replacive channel transposed orizontally. Indeed their melt focussing model at high T fails to reproduce half of the peridotites exposed at MOR.

A similar model can be applied to Penrose type ophiolites, although the Oman Drilling program revealed a much higher variability under the crust mantle boundary than a dunite zone. Eventually this model can be applied to fast spreading ridge, although site 895 shows a huge variability too. By no means this model can be applied to the mantle dredged at slow spreading ridges, more than half of MOR on our planet.

To what extent can this model be generalised? How a Nat Comm paper arguing that the abyssal mantle is stratified in such a manner will be taken by the community?

We did not drill so deep, and we can imagine the mantle being stratified in any possible way. But for what we see so far, the mantle is way more heterogeneous.

At this point I feel that my help as reviewer is completed. The authors take their responsibility for proposing this message, on which i do not agree. They will possibly make a provocative contribution, which as editor you can find suitable for the journal.

Best regards

Alessio Sanfilippo

Reviewer #2 (Remarks to the Author):

Review of manuscript#: NCOMMS-22-07539A, 'Vertical depletion of ophiolitic mantle reflects melt focusing and interaction in the asthenospheric column under oceanic spreading centers' by Qing Xiong et al.

I reviewed this manuscript twice. The new version is very clear, and in my opinion, could be suitable for publication. The geological and sample descriptions as well as the geochemical data are of high quality, and the proposed model/concept interesting for a broad audience working on ridge spreading processes. The few points needing to be addressed I mentioned in my last review have been taken into account, and the manuscript modified accordingly. Answers in the rebuttal letter are also relevant.

I also carefully read the two reviews by Dr. Alessio Sanfilippo. I agree with him that abyssal peridotites collected along slow spreading centres are extremely variable in terms of mineralogy and chemical compositions, variability not seen in the samples/data presented by Xiong and colleagues but observed in other ophiolites such as in Oman. On the other hand, abyssal peridotites outcrop along slow ridges only due to tectonism, either along transform faults or exhumed at OCC due to detachment faults development. These specific tectonic events initiate at HT magmatic stage and influence melt migration and melt/peridotite reaction, thus partly responsible for the evoked variability especially for refertilization processes (even if I agree, recent works show that part of the variability also results from ancient depletion processes). Peridotites sampled in the Pacific Ocean were also collected in a 'peculiar' geological context, at the Hess Deep tectonic window/propagating system.

For sure part of the variability observed in abyssal peridotites is inherited from old processes, but part also results from tectonism that led to their exposure on seafloor or along faults. So what in depth where the mantle has not been exhumed? Possibly the stratification proposed by Xiong and collaborators? Interesting proposition, a reason why I consider the manuscript well revised and improved compared to the first version. Of course one way to test such model would be to drill deeper in present-day ocean to reach more than 200m in depth, preferably not in areas governed by mantle exposure along faults but in unfaulted oceanic lithospheric portions if exist. Unfortunately due to technical limitations this goal will not be reached soon. Feedbacks between studies on present-day ridges and on ophiolites are necessary to better understand spreading processes, even if we get different informations according to specific local contexts. Should geologists avoid proposing models for spreading ridge processes, based on ophiolites? I don't think.

So in a way I agree both with Alessio Sanfilippo concerning its main concern, and with authors' responses in their rebuttal letter. However, I admit to definitely agree with reviewer Alessio Sanfilippo when he says :

'Eventually this model can be applied to fast spreading ridge, although site 895 shows a huge variability too (see my comment here above). By no means this model can be applied to the mantle dredged at slow spreading ridges (yes, but what in depth where no exhumation process occur? my comment above), more than half of MOR on our planet.'

In this view, as more than half of MOR are slow spreading, maybe authors should soften their conclusions (but losing provocative issues at the same time). I have no other comment than to recommend this interesting study for publication. I noticed only a very few points here below.

Best regards

Line 33: physicochemical

Line 35: 'in' spreading centers, under? or beneath rather than in?

Sample descriptions. Please indicate in Supplementary Fig. 2 caption how mineral modes have been calculated (using whole rocks and minerals major element compositions?)

Lines 125-127: 'In the CHZ and NHZ, the peridotites generally become more pyroxene-poor and Ol-rich from south to north, with gradual reduction in the grain sizes of both Opx and Cpx and enlargement of Ol and Sp grains (Supplementary Figs. 1e-1j, 2).'

It is not obvious in Supplementary Fig. 1. Picture (g) shows an orthopyroxene bigger than in pictures (i)-(m), and pictures selected to illustrate the NDZ show the smallest spinel grains.

Response letter

Responses to comments from Reviewer #1 (Dr. Alessio Sanfilippo)

Reviewer #1 (Remarks to the Author):

Dear Editor. I have read the new version of the manuscript with interest, following the response to my review as provided by the authors. I admit that they considered my concerns and modified some parts of the manuscript accordingly. In the response letter, they convincingly reply to the main arguments and provide a new revised version of the chemical model to account for the trace element composition of the rocks. On this basis, they state that the paper is now suitable for publication in your journal.

The main reason for my rejection in the first version is that, in my view, this paper does not provide a model applicable to the oceanic lithosphere, because the stratification that they see in the Lubosa ophiolite is not what we expect to be seen in the abyssal mantle. This argument is still there. They argue that abyssal peridotites represent a peculiar portion of the mantle, whereas km-thick oceanic sequences can only be studied at ophiolites. Still, the variability seen at MOR is not comparable to what provided here. The first 200 m of mantle of Luobusa ophiolites are mostly dunites, with some rare harzburgites or troctolites veins (although no analyses are provided). But dunites are rare at MOR, and if you plot the compositional variability of the dredged and drilled samples at present day ocean, it would cover entirely that of this 2 km-thick section. Even if "peculiar", the mantle manifests himself at MOR as an extremely variability assemblage of residual lithologies, variably interacted with melts at high and low T, at high at low P, locally refertilized, impregnated and intruded by gabbroic materials and veins with different composition. Such a variability is a first order process and indicates that refertilization, rather than extreme depletion, is a common characteristic of AP. And such a huge amount of refertilized lithologies are not seen in the Luobusa ophiolites and cannot survive in a huge vertical replacive channel transposed horizontally. Indeed their melt focusing model at high T fails to reproduce half of the peridotites exposed at MOR.

Our response: Thanks for the comments and suggestions. We agree that abyssal peridotites commonly record refertilization, because of melt metasomatism at low temperatures and within shallow lithospheric mantle along tectonic faults. Reviewer#2 also provides relevant comments and explanation below. The stratified mantle showing upward depletion proposed in this study should be located below the shallowest mantle from where abyssal peridotites are generally derived. Abyssal peridotites thus only sample the epidermis of oceanic lithospheric mantle exposed along faults in mid-ocean ridges and transform systems, and do not have vertical and large-scale spatial constraints as those given by well-preserved ophiolitic mantle. Our melt focusing and interaction model can therefore well explain the features of relatively deeper lithospheric mantle and can be complementary to the knowledge from abyssal peridotites.

A similar model can be applied to Penrose type ophiolites, although the Oman Drilling program revealed a much higher variability under the crust mantle boundary than a dunite zone. Eventually this model can be applied to fast spreading ridge, although site 895 shows a huge variability too. By no means this model can be applied to the mantle dredged at slow spreading ridges, more than half of MOR on our planet. To what extent can this model be generalised? How a Nat Comm paper arguing that the abyssal mantle is stratified in such a manner will be taken by the community?

Our response: This is a constructive suggestion. We have revised that our model can be applied to oceanic spreading centers with high melt fluxes, such as fast-spreading mid-ocean ridges and forearc/backarc centers. For slow-ultraslow spreading centers, there may be similar melt focusing and interaction in the asthenospheric upwelling column, but the thicker thermal boundary layer likely

suppresses the exposure of such upwards-depleted mantle and results in complex and strong metasomatic enrichments in the exposed lithospheric mantle (e.g., abyssal peridotites from such settings). We therefore believe that our model has general implications, at least for fast-spreading centers, and will be attractive for the community and large audience.

We did not drill so deep, and we can imagine the mantle being stratified in any possible way. But for what we see so far, the mantle is way more heterogeneous.

Our response: We expect that future oceanic drill holes can reach deeper mantle in un-faulted oceanic lithospheric regions, and can test our model. In addition, we admit that the asthenospheric source mantle is heterogeneous and the stratification of lithospheric mantle can be produced in any possible ways, but our model seems to be the most reasonable one to explain the vertical compositional variations of oceanic uppermost mantle.

At this point I feel that my help as reviewer is completed. The authors take their responsibility for proposing this message, on which i do not agree. They will possibly make a provocative contribution, which as editor you can find suitable for the journal.

Our response: Thanks for the help to improve this manuscript. We have revised the model to be applied to spreading centers with high melt fluxes, which we believe is still suitable for and meets the scope of Nature Communications.

Best regards

Alessio Sanfilippo

Responses to comments from Reviewer #2 (Anonymous)

Reviewer #2 (Remarks to the Author):

Review of manuscript#: NCOMMS-22-07539A, 'Vertical depletion of ophiolitic mantle reflects melt focusing and interaction in the asthenospheric column under oceanic spreading centers' by Qing Xiong et al.

I reviewed this manuscript twice. The new version is very clear, and in my opinion, could be suitable for publication. The geological and sample descriptions as well as the geochemical data are of high quality, and the proposed model/concept interesting for a broad audience working on ridge spreading processes. The few points needing to be addressed I mentioned in my last review have been taken into account, and the manuscript modified accordingly. Answers in the rebuttal letter are also relevant.

I also carefully read the two reviews by Dr. Alessio Sanfilippo. I agree with him that abyssal peridotites collected along slow spreading centres are extremely variable in terms of mineralogy and chemical compositions, variability not seen in the samples/data presented by Xiong and colleagues but observed in other ophiolites such as in Oman. On the other hand, abyssal peridotites outcrop along slow ridges only due to tectonism, either along transform faults or exhumed at OCC due to detachment faults development. These specific tectonic events initiate at HT magmatic stage and influence melt migration and melt/peridotite reaction, thus partly responsible for the evoked variability especially for refertilization processes (even if I agree, recent works show that part of the variability also results from ancient depletion processes). Peridotites sampled in the Pacific Ocean were also collected in a 'peculiar' geological context, at the Hess Deep tectonic window/propagating system.

For sure part of the variability observed in abyssal peridotites is inherited from old processes, but part also results from tectonism that led to their exposure on seafloor or along faults. So what in depth where the mantle has not been exhumed? Possibly the stratification proposed by Xiong and collaborators? Interesting proposition, a reason why I consider the manuscript well revised and improved compared to the first version. Of course one way to test such model would be to drill deeper in present-day ocean to reach more than 200m in depth, preferably not in areas governed by mantle exposure along faults but in unfaulted oceanic lithospheric portions if exist. Unfortunately due to technical limitations this goal will not be reached soon. Feedbacks between studies on present-day ridges and on ophiolites are necessary to better understand spreading processes, even if we get different informations according to specific local contexts. Should geologists avoid proposing models for spreading ridge processes, based on ophiolites? I don't think.

Our response: Thanks for these suggestions. We have clarified the implications in the revision (L318-347) to incorporate the above suggestions. Studying ophiolites and abyssal peridotites are not contradictory, but reveal complementary information to form a complete picture of oceanic spreading processes and the origin of oceanic lithosphere.

So in a way I agree both with Alessio Sanfilippo concerning its main concern, and with authors' responses in their rebuttal letter. However, I admit to definitely agree with reviewer Alessio Sanfilippo when he says: 'Eventually this model can be applied to fast spreading ridge, although site 895 shows a huge variability too (see my comment here above). By no means this model can be applied to the mantle dredged at slow spreading ridges (yes, but what in depth where no exhumation process occur? my comment above), more than half of MOR on our planet.'

In this view, as more than half of MOR are slow spreading, maybe authors should soften their conclusions (but losing provocative issues at the same time). I have no other comment than to recommend this interesting study for publication. I noticed only a very few points here below.

Our response: Thanks for this suggestion. We have revised our model that can be applied to oceanic spreading centers with high melt fluxes, such as fast-spreading mid-ocean ridges and forearc/backarc centers. For slow-ultraslow spreading centers, our model may also be suitable for explanation of the origin of deeper lithospheric mantle that is not refertilized at low temperatures.

Best regards

Line 33: physicochemical

Our response: Revised (L32 in the revision).

Line 35: 'in' spreading centers, under? or beneath rather than in?

Our response: Revised (L35 in the revision).

Sample descriptions. Please indicate in Supplementary Fig. 2 caption how mineral modes have been calculated (using whole rocks and minerals major element compositions?)

Our response: Revised (see the Supplementary Information file).

Lines 125-127: 'In the CHZ and NHZ, the peridotites generally become more pyroxene-poor and Ol-rich from south to north, with gradual reduction in the grain sizes of both Opx and Cpx and enlargement of Ol and Sp grains (Supplementary Figs. 1e-1j, 2)'. It is not obvious in Supplementary Fig. 1. Picture (g) shows an orthopyroxene bigger than in pictures (i)-(m), and pictures selected to illustrate the NDZ show the smallest spinel grains.

Our response: Revised accordingly (L127 in the revision and Supplementary Figure 1g).